

# Greenhouse gas and energy fluxes in a boreal peatland forest after clearcutting

Mika Korkiakoski[1], Juha-Pekka Tuovinen[1], Timo Penttilä[2], Sakari Sarkkola[2], Paavo Ojanen[3], Kari Minkkinen[3], Juuso Rainne[1], Tuomas Laurila[1], Annalea Lohila[1]

[1]Finnish Meteorological Institute, P.O. Box 503, FI-00101 Helsinki, Finland
[2]Natural Resources Institute Finland, Viikinkaari 4, 00790 Helsinki, Finland
[3]University of Helsinki, Department of Forest Sciences, P.O. Box 27, 00014 University of Helsinki, Finland

*Correspondence to*: Mika Korkiakoski (mika.korkiakoski@fmi.fi)

**Abstract.**

The most common forest management method in Fennoscandia is rotation forestry including clearcutting and forest regeneration. In clearcutting, stem wood is removed and the logging residues are either removed or left on site. Clearcutting changes the microclimate and vegetation structure at the site, both of which impact the site's carbon balance. Peat soils with poor aeration and high carbon (C) densities are especially prone to such changes, and significant changes in C stocks and greenhouse gas exchange can be expected. We measured carbon dioxide ($CO_2$) and energy fluxes with the eddy covariance

method for two years (April 2016 – March 2018) on a peatland drained for forestry. After the clearcutting, we observed a significant rise (23 cm) in the water table level. The site was also a large $CO_2$ source (first year: $3086 \pm 120$ g $CO_2$ m$^{-2}$ yr$^{-1}$, second year: $2072 \pm 141$ g $CO_2$ m$^{-2}$ yr$^{-1}$) after the clearcutting. These large $CO_2$ emissions resulted from the collapse of gross primary production (GPP) following the removal of photosynthesizing trees and the decline of ground vegetation. During the second summer (June–August) after the clearcutting, GPP had already increased by 96% and total ecosystem respiration

decreased by 14 % from the previous summer. As a result, net $CO_2$ emissions decreased during the second summer after clearcutting compared to the first one. The Bowen ratios were different in 2016 and 2017, starting at 2.6 in May 2016 and decreasing to less than 1.0 in August 2016, while in 2017 it varied mostly within 0.6−1.0. This was due to a 33% decrease in the sensible heat flux and a 40% increase in the latent heat flux from the 2016 values, probably due to the recovery of ground vegetation that increased evapotranspiration and albedo of the site. In addition to $CO_2$ and energy fluxes, we measured methane

($CH_4$) and nitrous oxide ($N_2O$) fluxes with manual chambers. After the clearcutting, the site turned from a small $CH_4$ sink into a small source and from $N_2O$-neutral to a significant $N_2O$ source. Compared to the large $CO_2$ emissions, the global warming potential (GWP$_{100}$) of the $CH_4$ emissions was negligible. Also, the GWP$_{100}$ due to increased $N_2O$ emissions was less than 10% of that of the $CO_2$ emission change.



## 1 Introduction

The northern peatlands cover approximately 3% of the Earth's land surface (Clarke and Rieley, 2010), most of which is located in the boreal region (Fischlin et al., 2007). The boreal and subarctic peatlands are substantial reservoirs of carbon (C), storing 270–550 Pg C in total (Turunen et al., 2002; Yu, 2011). These reservoirs are affected by peatland drainage, which has been a common practice in northern countries. In Finland, more than half of the original peatland area of 100 000 km$^2$ has been drained, mostly for forestry (Päivänen and Hånell, 2012). Drainage lowers water table level (WTL), which accelerates the peat decomposition rate due to the increased availability of oxygen. This often leads to higher carbon dioxide ($CO_2$) emissions from the soil (Maljanen et al., 2010; Ojanen et al., 2013). On the other hand, a well-performed drainage increases root aeration and nutrient availability, which enhances tree growth and $CO_2$ uptake by trees (Minkkinen et al., 2001). In addition, methane ($CH_4$) emissions decrease, and the soil may even turn into a net $CH_4$ sink if the site is well-drained (Maljanen et al., 2010; Ojanen et al., 2010). This results from the thickening of the oxic peat layer, which enhances $CH_4$ oxidation in the soil and from the disappearance of deep-rooted vascular plants, which in natural mires feed anaerobic microbes with fresh carbon and transport methane from catotelm to atmosphere through their aerenchyma. In contrast to $CH_4$, nitrogen oxide ($N_2O$) emissions often increase after drainage at nitrogen-rich minerotrophic peatlands (Maljanen et al., 2010; Ojanen et al., 2010, 2018) because nitrification (process that produces nitrate and $N_2O$ as a by-product) needs nitrogen and oxic conditions.

The most common method of forest management in Finland is rotation forestry including clearcutting and forest regeneration. In clearcutting, stem wood is removed, while the logging residues, i.e. foliage biomass, branches, stumps and roots, are either removed or left on site. After clearcutting, the site is prepared, e.g., by mounding or scalping, and a new tree generation is established by planting or sowing, or naturally from surrounding seed-trees. Removing the trees changes the local microclimate as more solar radiation reaches the soil surface, which increases soil temperature (Edwards and Ross-Todd, 1983; Londo et al., 1999) and its diel variation. This affects the carbon cycle, as higher soil temperature potentially increases soil respiration. In peatland forests, however, removing tree biomass raises the WTL by diminishing transpiration (Sarkkola et al., 2010) which in turn may decrease soil respiration and slow down the peat decomposition rate due to the reduced volume of aerated peat. On the other hand, photosynthesis is diminished due to the removal of trees and decline of ground vegetation. Mäkiranta et al. (2010) found that ground vegetation recovers rather fast in a peatland forest after clearcutting; however, after three years the recovery was still insufficient to compensate for the high ecosystem respiration produced by the large amount of fresh organic matter. The increase in ecosystem respiration is accounted for the increased soil organic matter decomposition under the logging residues and by the decay of these residues (Mäkiranta et al., 2012). As boreal forest grows slowly, it may take from 8 up to 20 years to turn the forest back to a net carbon sink (Fredeen et al., 2007; Kolari et al., 2004; Mäkiranta et al., 2010; Pypker and Fredeen, 2002; Rannik et al., 2002; Schulze et al., 1999).

The $CO_2$ and energy fluxes between ecosystems and the atmosphere are commonly measured with the eddy covariance (EC) method (Aubinet et al., 2012). Previous work concerning clearcutting in forests with mineral soil has involved both EC (Clark et al., 2004; Humphreys et al., 2005; Kolari et al., 2004; Kowalski et al., 2004, 2003; Machimura et al., 2005; Mamkin et al.,



2016; Takagi et al., 2009; Williams et al., 2013) and soil chambers (Howard et al., 2004; Londo et al., 1999; Zerva and Mencuccini, 2005). Also, soil chambers have been used on clear-cut peatland forests (Mäkiranta et al., 2010; Pearson et al., 2012), whereas EC measurements have been lacking. In addition, most of this peatland forest gas exchange research has concentrated on carbon dynamics and overlooked the possibly important variations in energy and water fluxes.

Exchange of sensible and latent heat constitutes an important part of the surface energy balance, driving many local, regional and global scale climatological processes. The available energy for these turbulent fluxes is determined predominantly by net radiation and the ground heat flux. The partitioning of available energy between the sensible and latent heat fluxes, which can be described in terms of the Bowen ratio, is strongly influenced by vegetation and soil properties (Betts et al., 2007). Also, release of latent heat to the atmosphere, i.e., evapotranspiration, is a key component of the water cycle in peatlands and forests,

also affecting soil moisture and forest productivity, which further affect the GHG fluxes in a forest. In a temperate deciduous broadleaf forest, evapotranspiration has been found to recover rapidly after clearcutting: the latent heat flux increased over the first three years, while the sensible heat flux declined correspondingly (Williams et al., 2013). It is unknown if this also happens in peatland forests.

In this study, we investigated the effect of clearcutting on GHG and energy fluxes and their environmental drivers in a boreal

peatland forest in southern Finland. The study was based on EC and soil chamber measurements performed during two years (April 2016 – March 2018) after the forest was clearcut. Our specific aims were:

1.   To estimate the magnitude of $CO_2$ fluxes and their environmental drivers after clearcutting.

2.   To quantify the development of surface heat fluxes after clearcutting.

3.   To investigate how soil $CH_4$ and $N_2O$ fluxes change due to clearcutting.

**2 Materials and methods**

**2.1 Measurement site**

The measurements were set up in a nutrient-rich peatland forest called Lettosuo, which is located in southern Finland (60°38' N, 23°57' E). The site was drained with widely-spaced, manually dug ditches probably during the 1930s, then drained more effectively in 1969 and fertilized with phosphorus and potassium. The distance between the ditches is on average 45 m, and

they were originally dug ca. 1 m deep but have since been partially filled with vegetation. After drainage and before clearcutting in 2016, the tree stand was dominated by Scots pine (*Pinus sylvestris*) with some pubescent birch (*Betula pubescens*). The understorey included mostly Norway spruce (*Picea abies*) and some small-sized pubescent birch. The tree stand was quite dense, which made the ground vegetation and moss layer patchy and variable due to irregular shading. Ground vegetation included herbs, such as *Trientalis europaea* and *Dryopteris carthusiana*, and dwarf shrubs, such as *Vaccinium myrtillus*

(Bhuiyan et al., 2017). The moss layer was dominated by *Pleurozium schreberi* and *Dicranum polysetum* with some *Sphagnum* mosses, such as *Sphagnum capillifolium*, *S. angustifolium* and *S. russowii*, in moist patches.



Clearcutting was performed within a trapezoidal area of 1.54 hectares (Fig. 1) between 29 February and 16 March 2016. After clearcutting, the logging residues were left at the site. The previous ground vegetation was almost totally destroyed by heavy machinery in the harvesting operation and the following drastic increase in solar radiation. In the following summer, some species adapted to the open, well-lighted conditions; for example, *Rubus idaeus*, *Carex canescens* and *Dryopteris carthusiana*

were observed here and there within the clear-cut area. Mounding was performed on 1–2 August 2016, and spruce (*Picea abies*) seedlings were planted in 2017. In addition to the clear-cut area, we had a similarly sized control area located south-west of the clear-cut area, where the forest was left in its original state with similar tree stand and vegetation composition as at the pre-harvest clear-cut area.

## 2.2 Measurement system

An EC system to measure turbulent $CO_2$ and energy fluxes was set up in the north-eastern part of the clearcut (Fig. 1), and the measurements started on 8 April 2016, approximately three weeks after the clearcutting had ended. The measurements continued until 7 April 2018. From this point on, the time periods of 8 April 2016 – 7 April 2017 and 8 April 2017 – 7 April 2018 are referred to as the first and second measurement year, respectively. The EC system included a three-axis sonic anemometer (uSonic-3 Scientific, METEK, Elmshorn, Germany) for wind speed and air temperature and a closed-path infrared

gas analyser (LI-7000, Licor Biosciences, Lincoln, NE, USA) for $CO_2$ and $H_2O$ mixing ratios. A sampling rate of 10 Hz was used for the EC system. The measurement height was 2.75 m, the flow rate was about 6 l min$^{-1}$, and the length of the inlet tube (inner diameter 3.1 mm, Bevaline IV) was 8 m. The mouth of the inlet tube was positioned 15 cm below the sonic anemometer. Air with a zero $CO_2$ concentration was used as the reference gas when calibrating the gas analyser. The micrometeorological sign convention is used throughout the paper: a positive flux indicates a flux from the ecosystem to the atmosphere (net

emission) and a negative flux indicates a flux from the atmosphere into the ecosystem (net uptake).

Auxiliary meteorological measurements were installed at the centre of the clearcut (80 m south of the EC mast) on 24 July 2015, i.e., before the clearcutting. A similar system was installed two weeks earlier for the control area (180 m south-east of the EC mast). Air temperature and relative humidity were measured at 2 m height (HMP155, Vaisala Corporation, Vantaa, Finland) and soil temperature profile at 5, 10, 20 and 30 cm depths (Pt100, Vaisala Corporation, Vantaa, Finland); the

measurements also included net radiation (NR Lite2 Net Radiometer, Kipp & Zonen, Delft, The Netherlands) and ground heat flux (HFP01, Hukseflux Thermal Sensors B.V., Delft, The Netherlands). In addition to these, global radiation (Pyranometer CMP3, Kipp & Zonen, Delft, The Netherlands) was measured from another EC mast above the canopy of the surrounding forest (250 m south from the clear-cut EC mast). The data were collected by data loggers (QML201C, Vaisala Corporation, Vantaa, Finland) as 30 min averages.

## 2.3 EC data processing

Half-hourly turbulent fluxes were calculated using standard EC methods (Aubinet et al., 2012). The 10 Hz raw data were block-averaged, and a double rotation of the coordinate system was applied (McMillen, 1988). The time lag between the gas



analyser and sonic anemometer signals was determined by cross-correlation analysis for each flux variable and 30 min period. Water vapour fluctuations affecting flux measurements with LI-7000 were compensated for (Webb et al., 1980); a corresponding compensation is not necessary for temperature (Rannik et al., 1997). The fluxes were corrected for systematic losses due to block averaging and attenuation of the highest frequencies in the cospectra between vertical wind speed and

mixing ratio; the transfer function method of Moore (1986) was used for this. For the high-frequency losses, the transfer functions describing the flux attenuation were determined separately for $CO_2$ and water vapour fluxes (with half-power frequencies of 1.3 and 0.6 Hz, respectively) using temperature cospectra as the reference. These functions were convoluted with generic cospectral distributions (Kaimal and Finnigan, 1994) to calculate the flux correction as a function of wind speed and atmospheric stability.

The 30 min averaged data were screened according to the following acceptance criteria: relative stationarity (Foken and Wichura, 1996) < 100%, internal LI-7000 pressure > 60 kPa, $CO_2$ mixing ratio > 350 ppm, number of spikes in the vertical wind speed and $CO_2$ concentration data < 150 of 18000. Also, the data from unsuitable wind directions and the periods of weak turbulence were discarded. For the latter, a friction velocity ($u_*$) limit of 0.125 m s$^{-1}$ was applied (Fig. S1). In addition, the footprint accumulated within the target area shown in Fig. 1 was required to exceed a limit of 0.75 to ensure that the

measured flux originated predominantly from the clear-cut area. The footprints were calculated using the model developed by Kormann and Meixner (2001) and input data measured with the sonic anemometer.

In addition to the vertical turbulent fluxes, the $CO_2$ fluxes associated with the storage of $CO_2$ below the measurement height were calculated from the change in the mean $CO_2$ concentration profile between consecutive half-hour periods. It was assumed that the $CO_2$ concentration in the air column from ground level to the measurement height of 2.75 m was constant.

There were only two periods when a measurement gap in the data was longer than five days: 30 September – 5 October 2016 and 28 April – 4 May 2017. After applying all the data filters described above, 30% of the 32635 half-hour periods recorded within the measurement period were accepted for further analysis (Fig. S2). The gap-filling procedure and uncertainty analysis of NEE are described in Appendix.

**2.4 Surface energy balance and Bowen ratio**

The surface energy balance can be expressed as

$$Q_H + Q_E = Q_N - Q_G - Q_S, \tag{1}$$

where $Q_H$ is sensible heat flux, $Q_E$ is latent heat flux, $Q_N$ is net radiation, $Q_G$ is the ground heat flux, and $Q_S$ is the sum of storage fluxes from other energy sinks and sources. In this study, we assumed that $Q_S = 0$.

The Bowen ratio, defined as

$$\beta = \frac{Q_H}{Q_E}, \tag{2}$$



is used to describe the partitioning of $Q_N - Q_G$ to $Q_H$ and $Q_E$ at the surface. When $\beta < 1$, more available energy at the surface is released to the atmosphere as latent heat than as sensible heat, and vice versa when $\beta > 1$. In this study, we present monthly Bowen ratios that were calculated from the monthly mean daytime (10:00–16:00, UTC+2) fluxes of $Q_H$ and $Q_E$. Gap-filling of the energy fluxes is described in Appendix.

## 2.5 Chamber measurements of GHG fluxes

For manual chamber measurements, a sampling transect was set up in the middle of the clear-cut area, where the GHG fluxes were measured between 29 June 2015 and 29 August 2017, mostly during the snow-free periods. The measurement interval varied between one week and one month, but there were longer gaps in autumn 2015 and spring 2016. The transect had two flux measurement plots at a distance of 4, 8, 12 and 22.5 m from the ditch, and at each distance there was an automatic WTL logger close to the flux measurement plots. Before starting the measurements, 2 cm deep grooves were carved into the soil surface for the chambers, and the grooves were occasionally renewed when necessary to keep the chamber sealing adequate. It should be noted that, even though logging residues were left at the site, the measurement plots did not have any above-ground residues. The fluxes were measured using a closed-chamber system with an opaque cylindrical chamber (height 30.5 cm, diameter 31.5 cm) including a mixing fan. A portable Gasmet DX4015 gas analyser (Gasmet Technologies Oy, Helsinki, Finland) based on Fourier transform infrared spectroscopy was used to measure $CO_2$, $CH_4$ and $N_2O$ mixing ratios every 5 s. The air was circulated in a loop between the gas analyser and the chamber, and the closure time was 10–11 min.

The fluxes were calculated the same way for all the gases (Korkiakoski et al., 2017). In short, both linear and exponential regression models were first fitted to the mixing ratio time series using the least-squares approach. The start and end points of the chamber closure were visually identified from the data for each closure. The first minute of each measurement was discarded to ensure that the sample air was properly mixed inside the chamber. After fitting, the mass flux ($F$) was calculated as

$$F = \left(\frac{dC(t)}{dt}\right)_{t=0} \frac{MPV}{RTA},$$ (3)

where $\left(\frac{dC(t)}{dt}\right)_{t=0}$ is the mixing ratio change in time determined from a linear or exponential model at the beginning of the closure, $M$ is the molecular mass of $CO_2$, $CH_4$ or $N_2O$ (44.01, 16.04 and 44.01 g mol$^{-1}$, respectively), $P$ is air pressure, $R$ is the universal gas constant (8.315 J mol$^{-1}$ K$^{-1}$), $T$ is the mean chamber headspace temperature during closure, and $V$ and $A$ are the volume and the base area of the chamber headspace, respectively. The snow depth and the height of mosses and other vegetation in the chamber headspace volume were taken into account, ignoring the pore space in the soil and snow. However, if the soil surface was frozen, the measurements were not made as it was not possible to properly seal the chamber. Finally, analyser-specific flux limits were determined to choose between the linear and exponential models (Korkiakoski et al., 2017). If the flux calculated with the linear model was smaller than the limit, then this estimate was considered more robust for the noisy data and adopted for the later analysis. These limits for the present system were 5.6 μg $CO_2$ m$^{-2}$ s$^{-1}$, 9.7 ng $CH_4$ m$^{-2}$ s$^{-1}$ and 12.5 ng $N_2O$ m$^{-2}$ s$^{-1}$.





The $CO_2$ flux measured with chambers represents forest floor respiration ($R_{ff}$), which is defined as the sum of heterotrophic and autotrophic respiration. As clearcutting affects soil processes, it should be noted that the $R_{ff}$ before and after clearcutting consist of different components. Before the clearcutting (the 2015 data), $R_{ff}$ includes ground vegetation and living roots of trees and ground vegetation, but after clearcutting (2016–2017) it instead includes the survived and regrown ground vegetation and living and dead roots.

The summertime sum of $R_{ff}$ was calculated for each summer separately by using Eq. A3 with hourly soil temperature at 5 cm depth. The uncertainty of this sum was estimated from the minimum and maximum sums based on the standard errors of model parameters ($R_0$ and $E$). The minimum and maximum sums were slightly asymmetrical around the mean sum, so the largest deviation from the mean sum was adopted as a conservative uncertainty estimate.

## 2.6 Statistical analysis

The statistical significance of the manual chamber gas flux differences between the years was tested with unpaired Student's t-test. Before using this test, the Shapiro–Wilk test was applied to test if the data were normally distributed. When at least one of the groups was not normally distributed, the Wilcoxon rank-sum test was used instead of Student's t-test. If the data sets were normally distributed, Levene's test was used to check the equality of variances, and if the variances were not equal, Welch's unequal variances t-test was used. The statistical tests were made with the Python programming language (Python Software Foundation, version 2.7, https://www.python.org) using SciPy (http://www.scipy.org/) library.

## 2.7 Water table level measurements and analysis

The water table levels within both the clear-cut area and non-managed control area were measured for one year before (2015) and two years after the clearcutting (2016–2017). Four automatic monitoring plots consisting of dipwells (perforated plastic tubes 120 cm long, 3.5 cm diameter) were set up at the centre of each area (Fig. 1), located at the distance of 4, 8, 12 and 22.5 m in a transect perpendicular to the ditch (ditch spacing was 45 m). In addition, in order to calibrate the automatic water table measurement data, manually monitored dipwells were installed close to the automatically monitored dipwells within the clear-cut and control areas. WTL was measured manually at weekly or fortnightly intervals during March–November. From the automated dipwells, WTL was recorded with an automatic probe (TruTrack WT-HR, Intech Instruments Ltd, Auckland, New Zealand; Odyssey Capacitance Water Level Logger, Dataflow Systems Limited, Christchurch, New Zealand) at hourly intervals. The recorded values were then calibrated with linear regression using the manually measured WTL data from both the control and clear-cut area.

The analysis of the clearcutting effects on WTL was based on the paired treatment approach (also called the calibration period – control area method) (e.g., Kaila et al., 2014; Laurén et al., 2009). We first calculated linear regressions between the WTL within the control and clear-cut areas for the pre-treatment period using the WTL logger data from 2015. Then we used this regression model and the post-treatment WTL data from the control area to predict WTL for the clear-cut area as if it had not



been harvested. The clear-cut effect was calculated as the difference between the calibrated post-clear-cut WTL measurements and the predicted background WTL values in the clear-cut area after the harvest.

## 3 Results

### 3.1 Meteorological and hydrological conditions

The mean annual air temperature at the site during the first and second measurement year were 5.6 °C and 4.4 °C, respectively. These temperatures were slightly higher and about the same, respectively, than the long-term (1981–2010) average at a nearby (Jokioinen, 35 km northwest of Lettosuo) weather station (4.6 °C; Pirinen et al., 2012). Both winters (DJF) were warmer (2016–2017: –3.0 °C, 2017–2018: –3.4 °C) than the long-term average (–5.3 °C), while the mean summer (JJA) temperature in 2016 (15.3 °C) was similar to and the summer 2017 1.6 °C cooler than the long-term average (15.2 °C). In addition,

compared to the precipitation at Jokioinen (long-term average 627 mm), the first measurement year was drier (502 mm) and the second year similar (656 mm). Especially autumn (SON) 2016 and winter 2016–2017 were much drier, while the springs and summers were quite similar to the long-term average conditions.

Both winters had a shallower snow cover than the long-term average annual maximum (28 cm; Pirinen et al., 2012). The maximum snow depths at Jokioinen were 12 cm and 22 cm in 2016–2017 and 2017–2018, respectively. Due to the especially

shallow snow cover in the first winter and the high temperatures in March, all the snow had melted by 23 March 2017. However, in 2018, the snow cover was still intact when the measurement period ended in 7 April.

The pre-harvest soil temperatures at 30 cm depth were similar within the clearcutting and control areas, but clearcutting increased their temporal variation. After clearcutting in late spring, the daily mean 30 cm soil temperature rose faster and was higher throughout the summer within the clearcut than at the control area (Fig. S3). Also, the 30 cm temperature within the

clearcut started to decline earlier in autumn and was lower during winter within the clearcut than the control area. The monthly mean diel variation of soil temperature at 5 cm depth ranged from 0.8 to 1.3 °C in July–September 2015 before clearcutting. After clearcutting, this variation increased to 2.3–2.9 °C and 2.1–3.7 °C in 2016 and 2017, respectively. On average the diel variation was 1.8 °C larger after the clearcutting than before it (Figs. 2b and S4). Increases in the diel temperature range were also observed at the 10 cm (1.2 °C) and 20 cm (0.33 °C) depths, but not anymore at 30 cm.

The mean WTL within the clear-cut area was about –36 cm during the pre-treatment growing season of 2015. After clearcutting, the WTL rose and remained continuously at a significantly higher level than the predicted background WTL. The largest post-treatment rise occurred in July–October when the average WTL was –23 cm and –24 cm in the clear-cut and –46 cm and –49 cm in the background model in 2016 and 2017, respectively (Fig. 3). As a consequence of heavy rain episodes in summer and autumn in 2017, the amplitude of WTL variations was larger than in the previous year (Fig. 3).



### 3.2 CO₂ exchange on an ecosystem level

#### 3.2.1 Seasonal and diel variations

When the measurements started at the beginning of April 2016, the daily mean NEE was mainly $< 0.08$ mg $CO_2$ m$^{-2}$ s$^{-1}$ but started to increase with temperature at the end of April, after the daily mean temperature exceeded 5 °C (Figs. 2a and 4). There

was a significant drop in NEE in the first half of June after a cold spell when the daily mean temperature decreased from 20 to 10 °C. By the end of June, NEE stabilized at around 0.21 mg $CO_2$ m$^{-2}$ s$^{-1}$. From the end of August, NEE decreased until reaching a stable level of 0.03 mg $CO_2$ m$^{-2}$ s$^{-1}$ in January 2017. At the end of March 2017, NEE started to gradually increase, and in mid-May it quickly rose to 0.08 mg $CO_2$ m$^{-2}$ s$^{-1}$. After this, NEE continued to increase, stabilizing at around 0.13 mg $CO_2$ m$^{-2}$ s$^{-1}$ in August and then decreasing from the end of September onwards.

Even though all the big trees were removed in clearcutting and most of the ground vegetation was destroyed, some sparse understorey was left at the site and weak photosynthesis could be observed simultaneously with the increased respiration in the end of April 2016 (Fig. 4). In May, the estimated magnitude of gross primary production (|GPP|, Appendix A) increased to 0.05 mg $CO_2$ m$^{-2}$ s$^{-1}$, which was about 25% of the respiration rate at that time. However, |GPP| decreased to 0.02 mg $CO_2$ m$^{-2}$ s$^{-1}$ in June, after which it started to increase again, reaching its maximum at 0.08 g $CO_2$ m$^{-2}$ s$^{-1}$ at the end of July and the

beginning of August. From that point on, photosynthesis started to weaken, marking the ending of the growing season, and ceased in mid-October, about a month before the first snow. In 2017, |GPP| increased from mid-May until July, when it stabilized at around 0.11 mg $CO_2$ m$^{-2}$ s$^{-1}$. Similarly to 2016, |GPP| decreased from August onwards.

The correlation between the night-time NEE (i.e., $R_{eco}$ in Eq. A3) and air temperature was stronger in summer 2017 than 2016 (Fig. 5). The model fits (Eq. A3) indicate that the temperature response of ecosystem respiration (parameter $E$) was also

stronger in 2017. However, the base respiration rate $R_0$ ($R_{eco}$ at 10 °C) was larger in the first than the second summer (0.199 vs. 0.173 mg $CO_2$ m$^{-2}$ s$^{-1}$). Like the temperature response of $R_{eco}$, the light response of GPP was stronger in summer 2017 than 2016 (Fig. 5). Also, the corresponding maximum gross photosynthesis rate ($GP_{max}$, Eq. A2) more than doubled (–0.226 vs. –0.100 mg $CO_2$ m$^{-2}$ s$^{-1}$).

The site was on average a $CO_2$ source throughout the day, also during the summer. It should be noted, however, that there were

25 still multiple half-hour periods, especially in August 2017, when the site acted as a $CO_2$ sink. A noticeable diel variation in NEE was observed mainly from April to October (Fig. 6). Between April and August, the mean diel NEE cycle had a minimum during the morning hours (5:00–10:00, UTC+2) and the highest emissions during the evening/night (21:00–1:00). After August, the lowest emissions took place later, at around noon. On the other hand, the highest emissions were observed earlier, at 20:00 in September and at 18:00 in October. In October, the diel cycle of NEE was still noticeable; it vanished in November

and reappeared in March 2017. In spite of the different mean fluxes, the amplitudes of their diel cycle were rather similar in 2016 and 2017, except from July to September when the amplitudes were much larger in 2017. Also, unlike in 2016, systematic diel variation was still obvious in November 2017. Even though such variation indicates significant photosynthesis during the midday hours, the monthly mean diel NEE cycle consistently showed positive fluxes.



### 3.2.3 CO₂ balances

The annual $CO_2$ balances of the first and the second year after clearcutting were $3086 \pm 98$ g $CO_2$ m$^{-2}$ ($\pm$ uncertainty; see Appendix) and $2072 \pm 115$ g $CO_2$ m$^{-2}$, respectively. About half of the annual $CO_2$ emissions during both measurement years took place during the summer months (JJA), totalling $1558 \pm 61$ g $CO_2$ m$^{-2}$ (Table 1, Fig. 7), and 73% ($2256 \pm 72$ g $CO_2$ m$^{-2}$)

of the annual total was accumulated during the May–September period. In 2017, however, these emissions decreased to $915 \pm 80$ g $CO_2$ m$^{-2}$ (June–August) and $1401 \pm 143$ g $CO_2$ m$^{-2}$ (May–September), i.e., by 41% and 38%, respectively. The effect of storage fluxes on the $CO_2$ balance was negligible as the daily mean storage flux typically varied within $\pm0.009$ mg $CO_2$ m$^{-2}$ s$^{-1}$ during summer and within $\pm0.004$ mg $CO_2$ m$^{-2}$ s$^{-1}$ during the other seasons.

The most significant period of $CO_2$ uptake was from June to September (Fig. 7). Considering only the summer months (JJA),

the integrated GPP was $-389$ and $-761$ g $CO_2$ m$^{-2}$ in 2016 and 2017, respectively (Table 1), i.e., the mean $CO_2$ uptake in the second summer after clearcutting increased by 96% from the first summer. On the other hand, the total summertime $R_{eco}$ decreased by 14%, from 1928 g $CO_2$ m$^{-2}$ in 2016 to 1652 g $CO_2$ m$^{-2}$ in 2017 (Table 1).

### 3.3 Soil GHG fluxes

The total pre-clearcut summertime (JJA) $R_{ff}$ was $1211 \pm 347$ g $CO_2$ m$^{-2}$ ($\pm$ uncertainty; see Sect. 2.5) and most of the measured

$CO_2$ fluxes varied from 0.05 to 0.25 g $CO_2$ m$^{-2}$ s$^{-1}$ (Fig. 8a). The $CH_4$ flux averaged over all the measurement plots was $-26 \pm 5$ ng $CH_4$ m$^{-2}$ s$^{-1}$ before the clearcutting. All the measured $CH_4$ fluxes were negative (Fig. 8b) ranging from $-103$ to $-2$ ng $CH_4$ m$^{-2}$ s$^{-1}$. $N_2O$ fluxes varied mostly from $-17$ to 33 ng $N_2O$ m$^{-2}$ s$^{-1}$ and averaged at $1 \pm 5$ ng $N_2O$ m$^{-2}$ s$^{-1}$ (Fig. 8c).

After the clearcutting, the total summertime $R_{ff}$ decreased to $619 \pm 184$ g $CO_2$ m$^{-2}$ in summer 2016 but partly recovered to $946 \pm 129$ g $CO_2$ m$^{-2}$ in 2017. Also, the temperature response of $R_{ff}$ got weaker and the $R_0$ parameter ($R_{ff}$ at 10 °C) decreased (Fig.

9). The strength of temperature response and $R_0$ partly recovered in 2017 but were still weaker than before the clearcutting.

$CH_4$ fluxes changed markedly after the clearcutting as the previously small mean $CH_4$ sink turned into a small $CH_4$ source in the first ($4 \pm 3$ ng $CH_4$ m$^{-2}$ s$^{-1}$) and second ($6 \pm 2$ ng $CH_4$ m$^{-2}$ s$^{-1}$) year after the clearcutting (Fig. 8b); the changes were significant for both years ($p < 0.001$ for 2016; $p < 0.001$ for 2017). However, the post-clearcut years were not significantly different from each other ($p = 0.893$). The post-clearcut fluxes varied mostly from $-14$ to 28 ng $CH_4$ m$^{-2}$ s$^{-1}$ with some

occasional emission peaks reaching up to 140 ng $CH_4$ m$^{-2}$ s$^{-1}$. Most of the measured fluxes were positive (emission), even though a single measurement plot could act both as a source and a sink on different measurement days.

Similarly to $CH_4$, the post-clearcut $N_2O$ fluxes were significantly higher than the pre-clearcut ones ($p = 0.002$ for 2016; $p < 0.001$ for 2017), and the post-clearcut years did not differ significantly from each other ($p = 0.778$). After the clearcutting, the mean annual fluxes were $228 \pm 26$ and $212 \pm 21$ ng $N_2O$ m$^{-2}$ s$^{-1}$ in 2016 and 2017, respectively (Fig. 8c). All the measurement

plots turned to large $N_2O$ sources, and the largest measured emission was 1339 ng $N_2O$ m$^{-2}$ s$^{-1}$. All the measured fluxes were positive, except for a plot furthest from the ditch that acted as a temporary $N_2O$ sink in 2017, especially in the June–July period.



### 3.4 Energy fluxes

The daily sums of sensible heat ($Q_H$) were about 1 MJ m$^{-2}$ day$^{-1}$ on average at the start of the EC measurement period in April 2016 and increased to a maximum of 4.7 MJ m$^{-2}$ day$^{-1}$ in June (Fig. 10). In July, the daily $Q_H$ sum was already decreasing with the decreasing net radiation ($Q_N$), and it turned negative in October. On the other hand, the daily sums of latent heat flux ($Q_E$)

5 were quite stable from April until August averaging at 1.0 ± 0.04 (± standard error of mean). In September, $Q_E$ started to decrease and reached zero at the end of October. The ground heat flux ($Q_G$) varied within 0–0.5 MJ m$^{-2}$ day$^{-1}$ from April to August, after which it varied mostly between –0.5 and 0 MJ m$^{-2}$ day$^{-1}$ until the end of April 2017.

In 2017, the seasonal dynamics of the energy fluxes were similar to 2016, but the magnitude of fluxes changed considerably. The summertime average daily sum of $Q_H$ decreased from 1.8 to 1.2 MJ m$^{-2}$ day$^{-1}$, while that of $Q_E$ increased from 1.3 to 2.3

10 MJ m$^{-2}$ day$^{-1}$. Also, the monthly sums of these fluxes changed significantly in the second year after the clearcutting. During the period of the highest fluxes (April–September), the $Q_H$ sum was 33% smaller in 2017 than 2016, while the $Q_E$ and $Q_N$ sums were higher by 40% and 13%, respectively. There was no significant change in $Q_G$.

The monthly mean midday $Q_H$ in May and June 2016 were 206 W m$^{-2}$ and 165 W m$^{-2}$, respectively (Fig. 11a). After June, the midday $Q_H$ started to decrease with decreasing $Q_N$, and the night-time $Q_H$ turned from zero to slightly negative. The midday

$Q_E$ also increased from April to June, reaching 88 W m$^{-2}$, and remained quite stable until September. The resulting monthly mean daytime Bowen ratio ($\beta$), increased from 1.7 in April to 2.6 in May 2016 and then declined to 0.8 in August. In 2017, the mean midday $Q_H$ was similar to that in 2016, except in July and September when the fluxes were 34% and 23% lower, respectively, than in 2016 (Fig. 11b). However, the daily maximum $Q_E$ was either similar to or higher (up to 48%) than $Q_H$ in all months except May 2017. Correspondingly, $\beta$ was lower than one (0.6–1.0) during the period of high fluxes in 2017, except

in May when it was 1.7.

### 4 Discussion

Dynamics of the CO$_2$ fluxes and flux components

At our site, the switch of the annual NEE from approximately zero to a large source, which took place after the clearcutting, can be explained by a decrease in the total |GPP| by about 88%, as estimated from the EC measurements in 2015 (Lohila,

unpublished data), following the removal of photosynthesizing trees and the decline of ground vegetation. |GPP| remained low and stable during the first summer after clearcutting, and the daytime NEE correlated only weakly with PPFD, reflecting the fact that neither vascular ground vegetation nor moss cover had developed significantly by that time. However, during the second summer, the total |GPP| increased by 96% from the first post-clearcut summer, and the increase in the coverage of ground vegetation was already noticeable (although not directly measured). This relatively fast recovery of ground vegetation

is similar to the leaf area index changes observed by Mäkiranta et al. (2010) after clearcutting in a nutrient-poor peatland forest.



The EC-based estimate of accumulated $R_{eco}$ decreased by 29% (40%) in the first (second) year after clearcutting from the 2015 value (Lohila, unpublished data), even though decomposing logging residues were left at the site and the summer following the clearcutting was slightly warmer than the one before it. In addition, the chamber measurements, which were not affected by aboveground logging residues, showed a 49% and 22% decrease in $R_{ff}$ after clearcutting in the first and second summer

after clearcutting, respectively. Possible reasons for this decrease include reduced peat decomposition due to raised WTL, ceased autotrophic respiration of (the harvested) trees roots and decreased autotrophic respiration of ground vegetation. The $CO_2$ flux from the decomposing belowground logging residues (roots) was not sufficiently large to compensate for this decrease.

After the removal of the canopy, more solar radiation reached the surface, heating it more during the daytime, while the heat

transfer from the soil to the atmosphere was enhanced during the night; together these resulted in a higher diel variation in soil temperature. On the other hand, the insulating impact of logging residues may have partly dampened the amplitude of the diel temperature cycle. In summer 2016, $R_{ff}$ was 32% of the $R_{eco}$ suggesting that two-thirds of the $R_{eco}$ during that time originated from the aboveground logging residues, which were not included in the chamber measurements of $R_{ff}$. However, in summer 2017, the total $R_{ff}$ increased by 35% and $R_{eco}$ decreased by 14%, indicating that a larger proportion (57%) of $R_{eco}$ originated

from the forest floor respiration than in the previous summer. This could be due to an increase in both autotrophic and heterotrophic respiration while the amount and the decomposition rate of the aboveground logging residues decreased (Mäkiranta et al., 2012). In addition, the NEE in summer 2017 was approximately equal to $R_{ff}$, which means that the increase in GPP was sufficient to balance the $CO_2$ emissions from the logging residues, but not from soil and plants.

The higher soil temperatures during the daytime may have enhanced peat respiration in the layer closest to the surface.

However, it has been shown in previous studies that the temperature response of respiration weakens when the soil gets dry, often resulting in lower respiration rates (Mäkiranta et al., 2009, 2010). Such weakening of the temperature response was also observed at Lettosuo. Moreover, the addition of logging residues may have increased $R_{eco}$ significantly, as demonstrated by Mäkiranta et al. (2012) who observed that the plots with logging residues had a $CO_2$ efflux twice as high as the plots without them. However, a later study by Ojanen et al. (2017) did not show a consistent increase in soil organic matter decomposition

rates. On the other hand, the soil decomposition rate was likely decreased by the raised WTL, which reduced the aerated peat volume, and drying of the almost bare soil surface, which was exposed to more solar radiation after clearcutting. Furthermore, the decline of ground vegetation after the clearcutting reduced autotrophic respiration, even though this process slightly recovered during the summer as the ground vegetation started to regrow.

In previous studies, a decrease has been observed in both $R_{eco}$ (Kowalski et al., 2004, 2003; Takagi et al., 2009) and $R_{ff}$

(Mäkiranta et al., 2010; Takagi et al., 2009; Zerva and Mencuccini, 2005) after clearcutting in peatland and upland forests. However, also increasing $R_{eco}$ and $R_{ff}$ have been reported (Kowalski et al., 2004; Londo et al., 1999). The differences between these results are most likely caused by a different effect of harvest on soil microbial activity, which is mostly controlled by soil temperature, moisture and nutrient availability (Davidson et al., 1998; Fontaine et al., 2004, 2007; Lloyd and Taylor,



1994). Other possible reasons for this variability include differences in, for example, root respiration, the type and amount of logging residue, and WTL, especially in peatland forests.

In peat soils, the decomposition of organic matter is mainly controlled by soil temperature and WTL (Blodau et al., 2004; Mäkiranta et al., 2009; Silvola et al., 1996), but soil temperature is typically the most influential factor explaining the temporal
dynamics in $R_{ff}$ (e.g. Mäkiranta et al., 2008; Ojanen et al., 2010). At Lettosuo, the correlation between the half-hourly $R_{eco}$ and soil temperature was mostly weak. This could be due to logging residues as their decomposition rate probably does not depend as much on soil temperature as on the temperature and moisture of the logging residues themselves. At times, especially when the temperature range was larger (in May and June), a significant relationship (correlation coefficient peaking at 0.57) was observed, and there was also an obvious temperature response that followed the annual cycle. Also, when WTL is closer (> −
15 cm) to the surface, it is usually a significant predictor of $R_{ff}$ in peat soils (Chimner and Cooper, 2003; Riutta et al., 2007). At Lettosuo after clearcutting, WTL varied mostly within 20–30 cm below the soil surface, and there was no correlation between $R_{eco}$ and WTL after removing the effect of soil temperature on $R_{eco}$ (data not shown). Because $R_{eco}$ combines many respiration components in addition to $R_{ff}$, it is likely that the high $CO_2$ efflux from logging residues, for example, masks the possible WTL effect on $R_{ff}$.

Our results show that after clearcutting a peatland forest becomes a large source of $CO_2$ to the atmosphere. Growing nutrient-rich peatland forests are commonly $CO_2$ neutral or small sinks (Meyer et al., 2013; Ojanen et al., 2013; Uri et al., 2017). Lettosuo was also about $CO_2$ neutral before the clearcutting (Lohila, unpublished data), so the change to a source of 3086 ± 120 g $CO_2$ m$^{-2}$ yr$^{-1}$ in the first year after clearcutting is substantial. Mäkiranta et al. (2010) measured $CO_2$ exchange in a clear-cut nutrient-poor peatland forest with soil chambers during a growing season (May–October) and reported slightly smaller
NEE (1990 g $CO_2$ m$^{-2}$ season$^{-1}$) than the NEE we measured at Lettosuo during a similar timeframe (2340 g $CO_2$ m$^{-2}$ season$^{-1}$). Among mineral soil forests, on the other hand, the annual $CO_2$ exchange measured with the EC method right after the clearcutting varies a lot; the annual balances ranging from ca. 400 to 4700 g $CO_2$ m$^{-2}$ yr$^{-1}$ (Amiro et al., 2010; Clark et al., 2004; Humphreys et al., 2005; Kowalski et al., 2004, 2003; Takagi et al., 2009; Williams et al., 2013). If considering only mineral soil forests, the forests in warmer climates typically emit more carbon after the harvest, but they also recover faster
than the boreal forests (Amiro et al., 2010). Compared to the studies cited above, the clear-cut area at Lettosuo was the second largest source of $CO_2$ immediately after clearcutting, being second only to the Slash pine plantation in Florida (Clark et al., 2004). The high $CO_2$ emission at Lettosuo as compared to other clearcut sites is apparently due to the decomposition of the peat soil in this nutrient-rich forest.

### Soil $CH_4$ and $N_2O$ fluxes

The $CH_4$ flux in peat soils is mostly controlled by WTL (Martikainen et al., 1995; Roulet et al., 1993), which divides the peat column into anoxic and oxic layers in which $CH_4$ production and oxidation occur, respectively. After clearcutting, the WTL rose from the −36 cm in 2015 to −23 cm in 2016, and the clear-cut was estimated to raise WTL by 23 cm. This made the near-surface oxic layer much thinner and the conditions for $CH_4$ oxidation less favourable, allowing a smaller amount of the $CH_4$



created in the anoxic layer to be oxidized before reaching the atmosphere. The WTL rise was enough to turn the site from a $CH_4$ sink into a small $CH_4$ source. Logging residues do not affect the $CH_4$ fluxes (Mäkiranta et al., 2012), but as was the case with $R_{ff}$, it is possible that the dryness of the year 2016 reduced the $CH_4$ emissions after the harvest. A similar turn of a clear-cut site from a $CH_4$ sink to a source has previously been found for both peat (Zerva and Mencuccini, 2005) and mineral soils

(Sundqvist et al., 2014). However, Huttunen et al. (2003) measured mostly $CH_4$ sinks in two drained peatland forests in southern Finland also after clearcutting, and the difference between the control and clear-cut sites was not statistically significant at either site.

The $N_2O$ fluxes were highly variable both before and after clearcutting. However, there was a strong increase from the mean pre-clearcut average flux of $1 \pm 5$ ng $N_2O$ m$^{-2}$ s$^{-1}$ to the post-clearcut flux of $228 \pm 26$ ng $N_2O$ m$^{-2}$ s$^{-1}$. Increases of $N_2O$

emissions after clearcutting have also been observed in peat and mineral soil forests after clearcutting by Huttunen et al. (2003) and Saari et al. (2009), but the fluxes at these sites were an order of magnitude smaller than at Lettosuo. It should be noted that the $N_2O$ measurements in this study do not include any above-ground logging residues, which are expected to raise $N_2O$ emissions further; Mäkiranta et al. (2012) found about three times as large $N_2O$ emissions from plots with logging residues as from plots without them.

Compared to the change in NEE at the site, the change in $CH_4$ emissions is negligible when considering its climatic impact in terms of the global warming potential (GWP) (IPCC, 2013). Even though the change in $N_2O$ fluxes to the present level at our site was large, the climatic effect of $N_2O$ was only about 10% of that due to the even larger change in NEE, when considering GWP over a period of up to 100 years (IPCC, 2013).

Energy fluxes

The mean daily $Q_E$ at Lettosuo during spring and summer right after the clearcutting varied mostly within 0.5–2.5 MJ m$^{-2}$ day$^{-1}$ and within 1.0–4.0 MJ m$^{-2}$ day$^{-1}$ during the second spring and summer. Compared to the values of 2–10 MJ m$^{-2}$ day$^{-1}$ measured in an upland forest in Western Russia (Mamkin et al., 2016), our numbers are low. In the case of $Q_H$, the fluxes at Lettosuo were systematically lower than those reported by Mamkin et al. (2016), but the difference was not as large as for $Q_E$. At Lettosuo, $\beta$ was lower in the second summer after the clearcutting as compared to the first summer; in summer 2016 $\beta$ was >1

most of the time, whereas in summer 2017 it was <1 for the whole summer. Thus the pattern was somewhat different from the results of Mamkin et al. (2016) who reported that $\beta < 1$ already throughout the first summer after the clearcutting. One reason for the smaller $Q_E$ at Lettosuo could be the drier-than-normal summers in 2016 and 2017 and the drier surface layer of the peat. In summer 2017 at Lettosuo, $Q_H$ decreased slightly and $Q_E$ doubled compared to 2016, suggesting a recovery of transpiration due to the gradual recovery of ground vegetation. This is in agreement with the results obtained by Williams et al. (2013) for

a clear-cut Norway spruce plantation in north-east USA, where a recovery of evapotranspiration and a decrease in $Q_H$ were observed across the years after the clearcutting. Moreover, the recovered ground vegetation at Lettosuo most likely increased albedo and thus prevented the soil from heating as much as in 2016, resulting in a smaller $Q_H$ over the whole summer. The clear-cut area at Lettosuo is small, but a larger scale clearcutting would probably affect local and regional climate due to





changes in surface energy balance fluxes. This is because $Q_H$ and $Q_E$ affect the properties and growth rate of the planetary boundary layer, influencing convection and the long-range transport of heat and humidity.

## 5 Conclusions

Based on our new and previous measurements, we conclude that clearcutting a peatland forest turns it into a large $CO_2$ source.
This results from the removal of photosynthesizing trees and the decline of ground vegetation and understorey. At Lettosuo, clearcutting caused an 86% decrease in the annual GPP, whereas $R_{eco}$ decreased by 20%. Removing the trees decreased transpiration, which caused the WTL to rise by 23 cm, which in turn likely decreased the peat decomposition rate due to the decreased volume of aerobic peat. Plant respiration also decreased as the plants were removed or destroyed. On the other hand, decomposition of logging residues likely increased $CO_2$ emissions from the site. In the second summer after the clearcutting,
ground vegetation and GPP recovered noticeably and $R_{eco}$ decreased. This reduced the net $CO_2$ emissions of the site by 41% compared to the first summer.

The soil turned from a small $CH_4$ sink into a small source after clearcutting due to the WTL rise. However, the radiative forcing related to this change was insignificant compared to that due to the change in NEE. Clearcutting turned the soil into a significant source of $N_2O$. This change produced a 100 yr GWP of about 10% of that due to the increased NEE at the site.
The mean daytime latent heat flux doubled from the first to the second year after clear-cutting, suggesting that transpiration through ground vegetation had significantly recovered. The recovered ground vegetation most likely also increased albedo and thus prevented the soil from heating as much as in 2016, which resulted in a smaller sensible heat flux.

Overall, the results in this study show that clearcutting peatland forests exerts a strong climatic warming effect through accelerated emissions of greenhouse gases. However, this study only demonstrates a short-term impact of two years, and more
extensive measurements are required to gain knowledge of the long-term effects of clearcutting in peatland forests.



## Appendix A: Gap-filling CO₂ data

To calculate seasonal and annual $CO_2$ balances, full time series are needed and thus any gaps in the $CO_2$ flux data need to be filled. For this, and to analyse the components of the carbon balance, we partitioned the measured $CO_2$ flux, i.e., net ecosystem exchange (NEE), into gross primary production (GPP) and total ecosystem respiration ($R_{eco}$):

$$\text{NEE} = \text{GPP} + R_{eco},\tag{A1}$$

For gap-filling, we expressed GPP as a function of irradiance (e.g., Aurela et al., 2015):

$$\text{GPP} = \frac{\alpha \times \text{PPFD} \times GP_{max}}{\alpha \times \text{PPFD} + GP_{max}},\tag{A2}$$

where PPFD is the photosynthetic photon flux density, $\alpha$ is the initial slope of the NEE response to PPFD, and $GP_{max}$ is the asymptotic gross photosynthesis rate in optimal light conditions. $R_{eco}$ was assumed to follow the Arrhenius-type model

described by Lloyd and Taylor (1994):

$$R_{eco} = R_0 \times \exp\left[E\left(\frac{1}{T_0} - \frac{1}{T_{air} - T_1}\right)\right],\tag{A3}$$

where $R_0$ is the ecosystem respiration at 10 °C, $E$ is the temperature sensitivity of the respiration, $T_{air}$ is the measured air temperature, $T_0 = 56.02$ K and $T_1 = 227.13$ K.

The parameters $E$ and $R_0$ were defined from the night-time data (PPFD < 20 µmol m$^{-2}$ s$^{-1}$) in two parts. First, the parameter $E$,

which was allowed to vary within 200–500 K, was determined with a 15 day moving window for each day, with the minimum number of observations set to 12. If there were not enough observations within a time window, then the window size was increased by one day both in the beginning and the end until enough data were found. The resulting window size varied between 15 and 27 days, but only 22 gap-filled days had a longer window than 15 days. Next, all the $E$ values that hit the allowed boundary values (200 and 500 K) were discarded and filled with 14 day moving medians. Finally, a 15 day moving

window, similar to the one in the first part, was used to determine $R_0$ by using the fixed $E$ values. Also, using the same moving window, $GP_{max}$ and $\alpha$ were determined from the daytime data. However, from 1 November 2016 until 8 March 2017 with no significant $CO_2$ uptake, a 5 day moving average was used to fill the gaps in the measured NEE.

All the calculations and analyses were made with the Python programming language (Python Software Foundation, version 2.7, https://www.python.org). For the fits, the least-squares method was used through the "polyfit" function of NumPy

(http://www.numpy.org/) library for the linear regression and the "curve_fit" function of SciPy (http://www.scipy.org/) library for the nonlinear fits.

## Appendix B: Uncertainty analysis of NEE

The $CO_2$ balance obtained from EC measurements has multiple potential error sources due to instrumental, statistical and methodological uncertainties. We included the most significant, although not all, random error sources. The random error





including the statistical measurement error ($E_{\text{meas}}$) inherent in EC measurements and the error caused by gap filling of missing data ($E_{\text{gap}}$) was estimated as follows (Räsänen et al., 2017):

$$E_{\text{meas/gap}} = \sqrt{\sum_i \frac{\left(\text{NEE}_{i,\text{obs}} - \text{NEE}_{i,\text{mod}}\right)^2}{n_{\text{obs/mod}}}} \sqrt{n_{meas/gap}}, \tag{B1}$$

where $\text{NEE}_{\text{obs}}$ is the 30 min flux that passed all the filtering procedures and $\text{NEE}_{\text{mod}}$ is the corresponding fitted NEE (Eqs. A1–

A3), and $n_{\text{meas/gap}}$ the number of the measured or gap-filled data. This method provides a conservative estimate for $E_{meas}$ (Aurela et al., 2002) and for $E_{\text{gap}}$ includes the effect of random variability on the model fits.

The annual systematic error caused by the friction velocity filtering ($E_{\text{ustar}}$) was estimated by recalculating the annual balance with modified data sets that were screened with two different $u_*$ limits (0.075 and 0.175 m s$^{-1}$). $E_{\text{ustar}}$ was calculated as the average difference between the annual balance calculated with the optimal $u_*$ threshold (0.125 m s$^{-1}$) and with the annual

balances calculated with the modified $u_*$ limits. Similarly, two additional different footprint limits (0.65 and 0.85) were adopted to estimate the annual error caused by the footprint filtering ($E_{\text{fp}}$).

The total uncertainty of the annual balance ($E_{\text{tot}}$) was calculated with the standard error propagation principle:

$$E_{\text{tot}} = \sqrt{E_{\text{meas}}^2 + E_{\text{gap}}^2 + E_{\text{ustar}}^2 + E_{\text{fp}}^2}, \tag{B2}$$

**Appendix C: Gap-filling energy fluxes**

Energy fluxes were gap-filled in several steps following the procedure described by Kowalski et al. (2003). First, the gaps in daytime $Q_{\text{H}}$ ($Q_{\text{N}} > 0$) were filled with monthly linear regressions with net radiation. Next, the night-time ($R_{\text{n}} < 0$) gaps in $Q_{\text{H}}$ were replaced by the corresponding $Q_{\text{N}}$. Finally, the daytime gaps in $Q_{\text{E}}$ were filled in such a way that the monthly mean energy balance closure was achieved, while during the night the missing $Q_{\text{E}}$ data were set to 0.



*Data availability*. The measured flux and meteorological data are available in Zenodo (https://zenodo.org/) research data depository.

*Competing interests*. The authors declare that they have no conflict of interest.

*Acknowledgements*. We are grateful for the financial support from the Maj and Tor Nessling foundation and from the Ministry of Transport and Communications through the Integrated Carbon Observing System (ICOS) research.



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



**Table 1. Annual and summertime (JJA) balances of net ecosystem exchange (NEE), modelled gross primary production (GPP) and modelled total ecosystem respiration ($R_{tot}$), and the modelled summertime sums of forest floor respiration ($R_{ff}$).**

| | Annual | | Summer (JJA) | |
|---|---|---|---|---|
| | Apr 2016 - Mar 2017 | Apr 2017 - Mar 2018 | 2016 | 2017 |
| NEE [g $CO_2$ m$^{-2}$] | 3086±98 | 2072±115 | 1558±61 | 915±80 |
| GPP [g $CO_2$ m$^{-2}$] | -659 | -1106 | -389 | -761 |
| $R_{eco}$ [g $CO_2$ m$^{-2}$] | 3727 | 3135 | 1928 | 1652 |
| $R_{ff}$ [g $CO_2$ m$^{-2}$] | - | - | 619 | 946 |





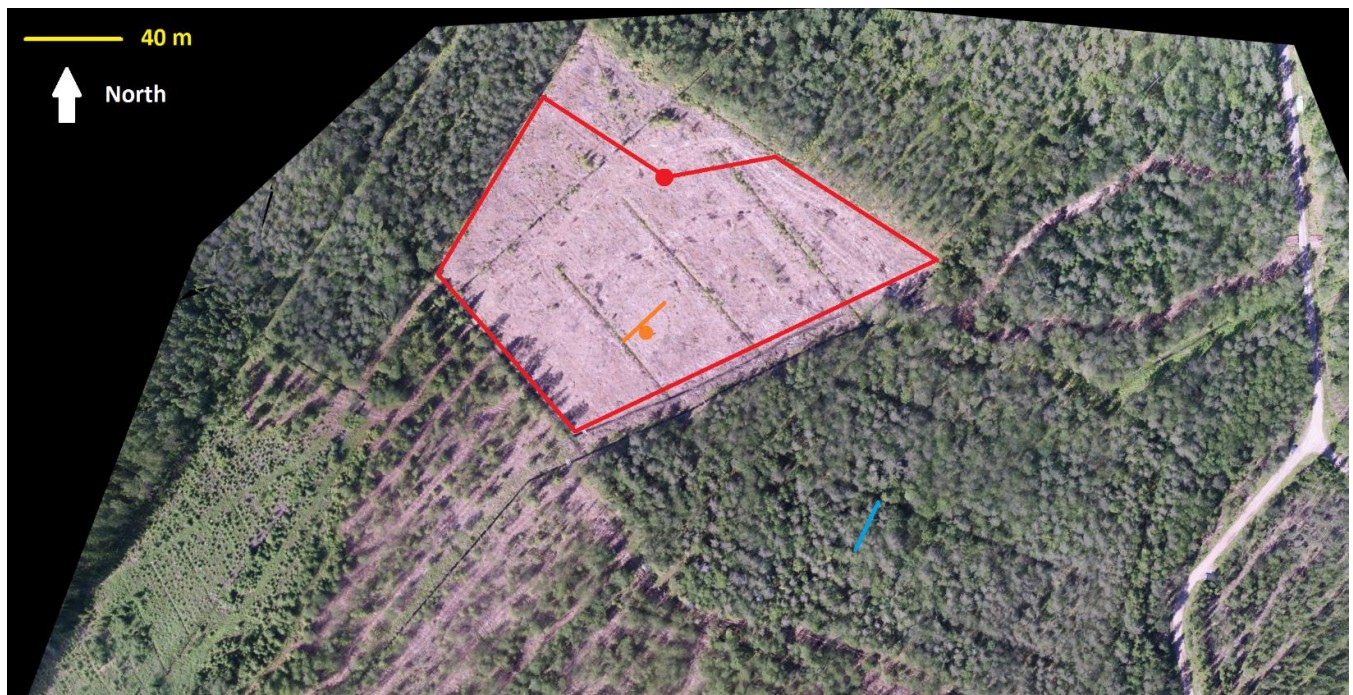

**Figure 1. Aerial view of the clear-cut site and the surrounding forest at Lettosuo. The red dot shows the location of the eddy covariance mast, and the red lines surround the target area (80° – 315°). The meteorological measurements were conducted at the location indicated by the orange dot, while the orange line marks the water table, soil temperature and chamber measurements (4, 8, 12 and 22.5 m from the ditch). Similar measurements of water table and soil temperature within the control site are shown with a light blue line.**





**Figure 2. Time series of (a) daily mean air temperature ($T_{air}$) and (b) hourly mean soil temperatures at 5, 10, 20 and 30 cm depths ($T_{soil}$) measured at Lettosuo, and (c) daily precipitation sum and (d) daily mean snow level recorded at the Jokioinen observatory (35 km northwest of Lettosuo). The clearcutting (red vertical line) was carried out in February–March 2016.**





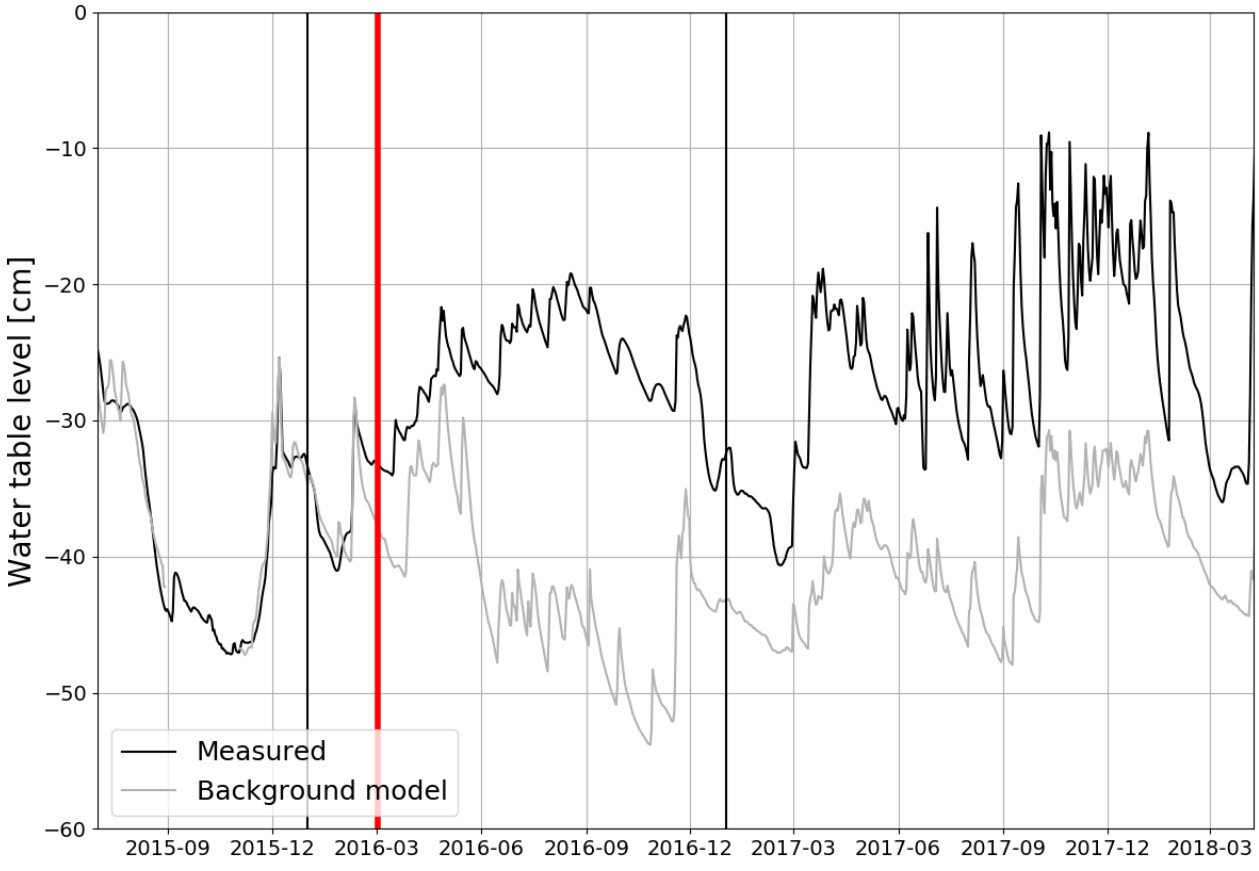

**Figure 3. Mean daily water table level in the clear-cut site with (measured; black line) and without (estimated; grey line) the harvesting effect averaged over all the measurement points (4, 8, 12 and 22.5 m from the ditch) from June 2015 to October 2017. The clearcutting (red vertical line) was carried out in February–March 2016.**

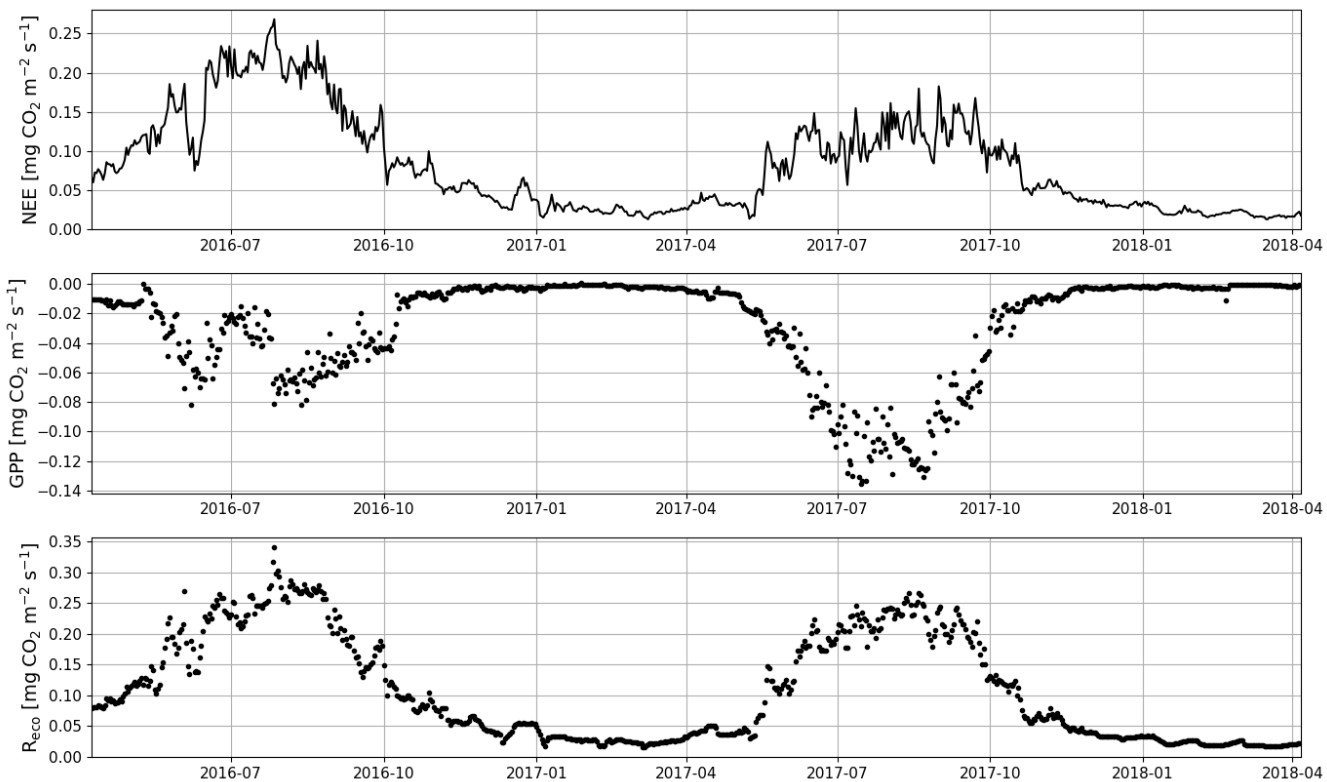

**Figure 4. The gap-filled time series of daily mean net ecosystem exchange (NEE, top panel) and the corresponding modelled gross primary production (GPP, middle panel) and ecosystem respiration ($R_{eco}$, bottom panel) based on Eq. (A2) and Eq. (A3), respectively.**



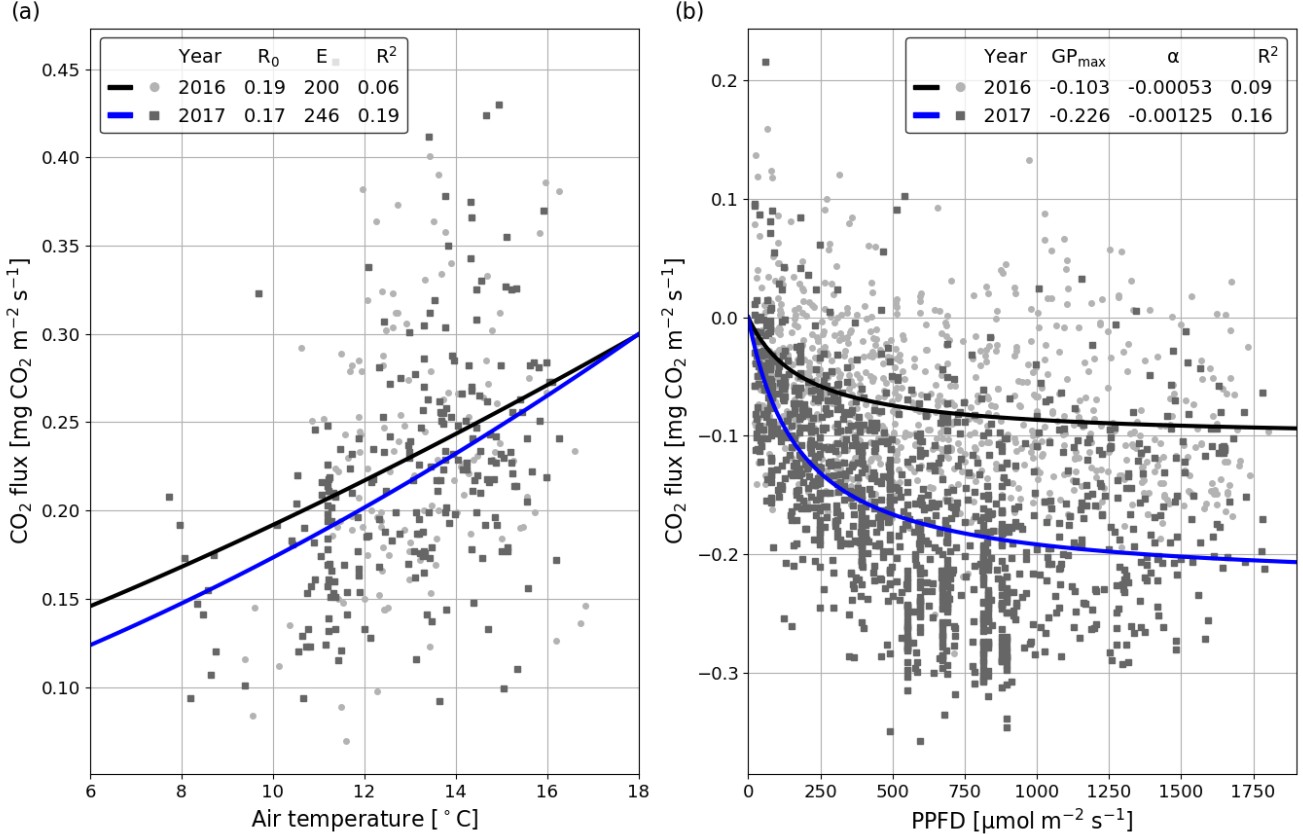

**Figure 5. Temperature (a; Eq. A3) and light responses (b; Eq. A2) of 30 min CO₂ fluxes for summers (JJA) 2016 (black) and 2017 (blue). The temperature response equation was fitted to the night-time data (PPFD < 20 μmol m⁻² s⁻¹) and the light response equation to the daytime (PPFD > 20 μmol m⁻² s⁻¹) data.**





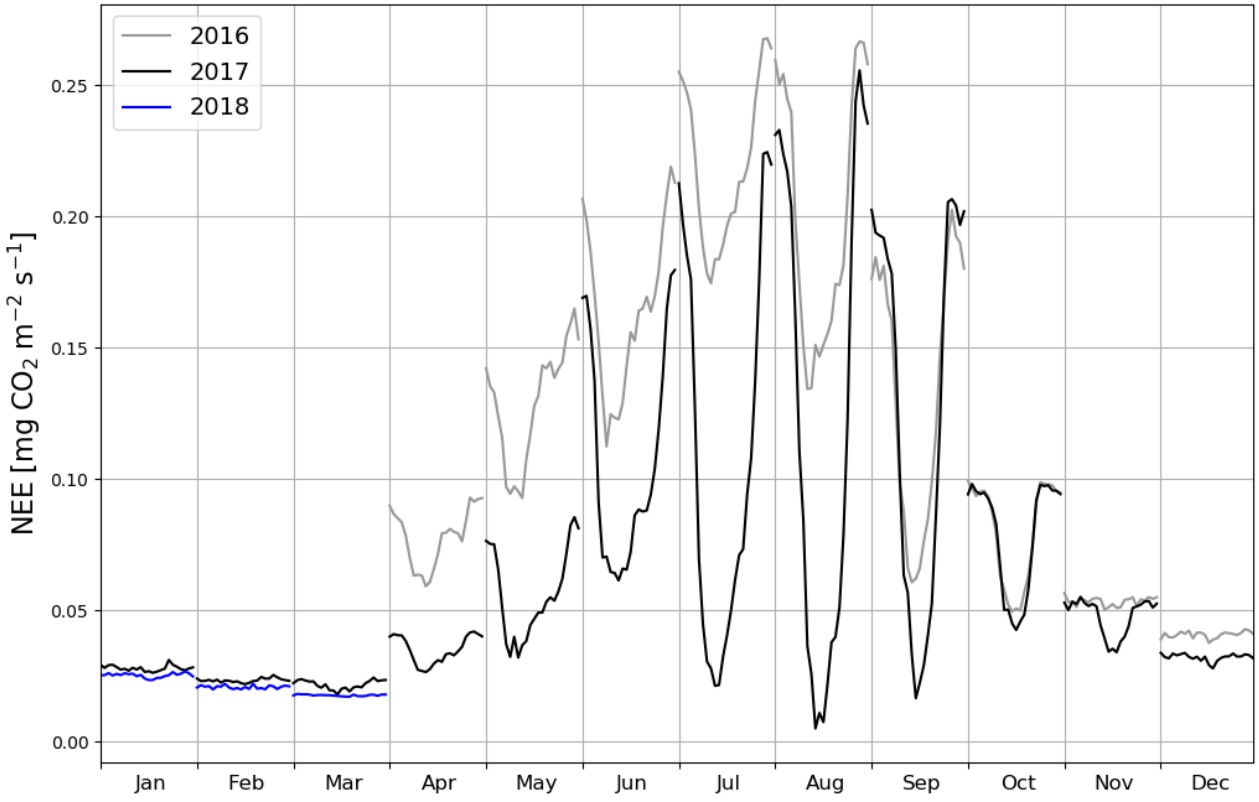

**Figure 6. Monthly mean diel cycles of gap-filled net ecosystem exchange (NEE) measurements in the first (2016, grey), second (2017, black) and third (2018, blue) calendar year after the clearcutting.**



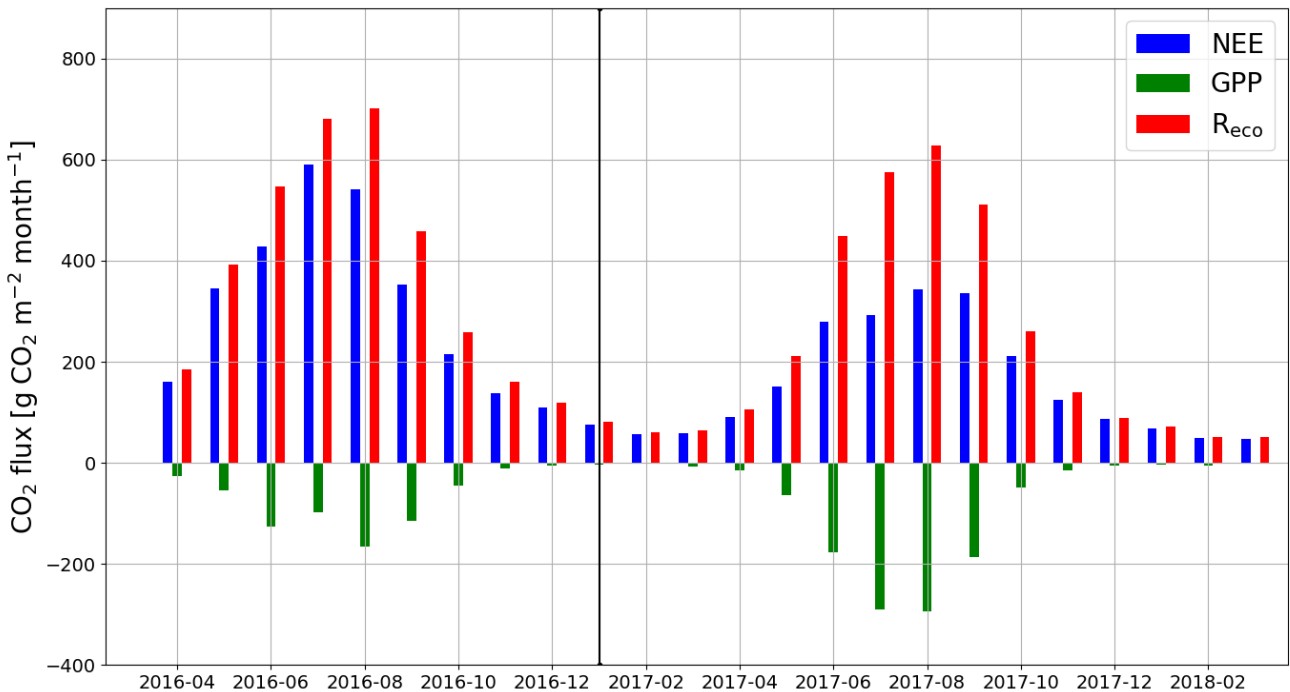

**Figure 7. The monthly sums of net ecosystem exchange (NEE), gross primary production (GPP) and ecosystem respiration ($R_{eco}$) based on EC data in April 2016 – March 2018.**





**Figure 8. Hourly mean fluxes measured with manual chambers of (a) CO₂, (b) CH₄ and (c) N₂O averaged over all measurement points (4, 8, 12 and 22.5 m from the ditch) from June 2015 to August 2017. The error bars show the standard error of the mean.**



**Figure 9. Temperature response (Eq. A3) of the hourly mean CO₂ fluxes measured with manual chambers in 2015 (black), 2016 (blue) and 2017 (green).**





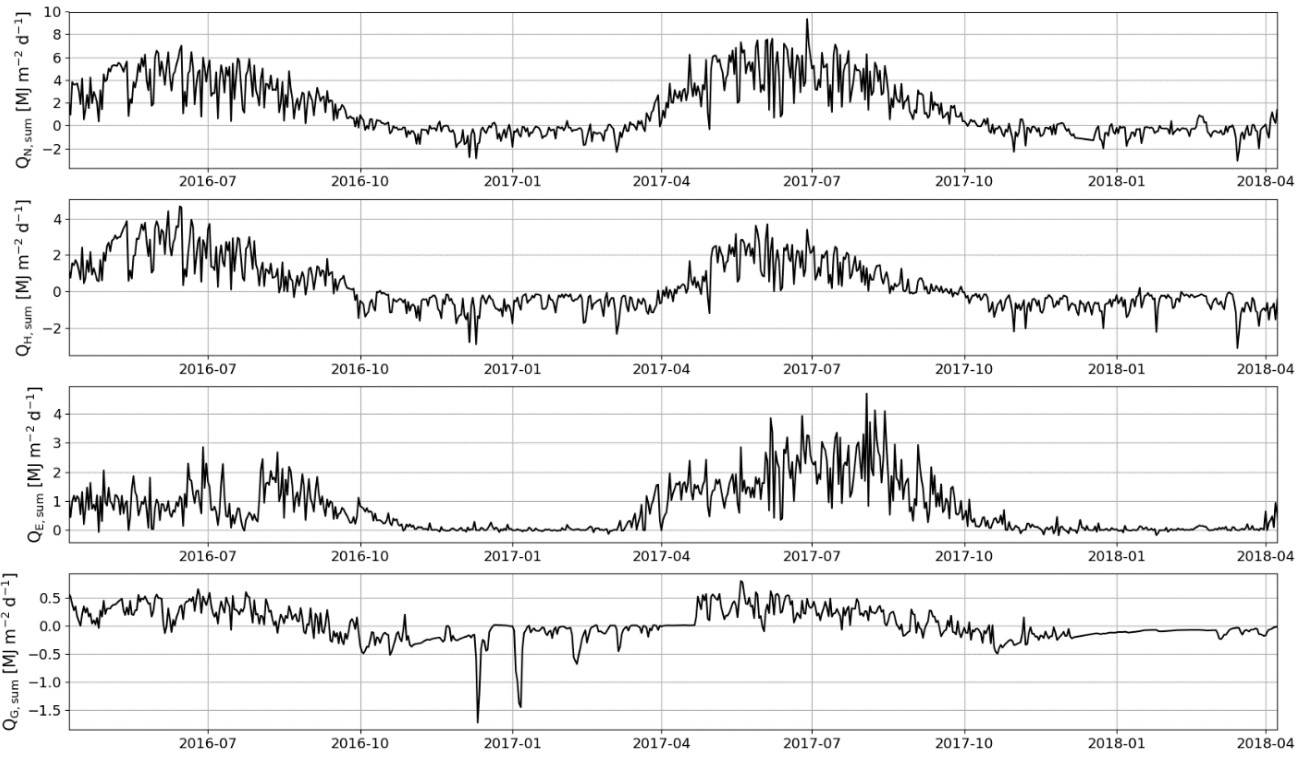

**Figure 10. Daily sums of net radiation ($Q_N$), sensible heat flux ($Q_H$), latent heat flux ($Q_E$) and ground heat flux ($Q_G$).**

...

...

...

...

...

...

...

...

...

...

...



**Figure 11. Monthly mean diel cycles of net radiation ($Q_N$), air temperature ($T_{air}$), sensible heat flux ($Q_H$), latent heat flux ($Q_E$) and ground heat flux ($Q_G$) and the monthly mean daytime (10:00–16:00, UTC+2) Bowen ratios ($\beta$) from April to September in 2016 (a) and 2017 (b).**