# Peer review of "Greenhouse gas and energy fluxes in a boreal peatland forest after clearcutting"

_Biogeosciences, 2018_

## Referee Comment (RC1) · Anonymous Referee #1 · 15 Jan 2019

General Comments: The study investigated all three main GHG gases ($CO_2$, $CH_4$ and $N_2O$) and heat fluxes after a clear cut on a moderately drained boreal organic soil. The authors used a combination of eddy-covariance and chamber method for flux determination. Studies determining the effect on GHG after clear cut exits for mineral soil, but not for organic soil. Details are nicely discussed.

My main concern is the experimental design of the study. The paper is based on two years of $CO_2$ measurement after the clear cut. Reference data (fluxes before the clear cut) are missing, although it is stated in the paper, that one year measurement of $CO_2$ fluxes before the clear cut exits. These data are not published (p.11 ll.24-25). However the main conclusion (clearcutting turned boreal forest from neutral to C source) is based on these unavailable data. In view of the generally large interannual
variability of carbon balances on organic soil, the main conclusion is weakly supported. Interannual variability is not even discussed within the paper, although the first year after clear cutting is drier and warmer than the following. Without a control site including GHG measurements it is difficult to figure out the influence of the clear cut and "normal" annual variability. It exits a control plot of a reference site, but only data of water table and soil temperature is included. There is a second EC tower, measuring above the canopy (p.4 l.27). It is not clear whether GHG flux data exits and could be used as reference. There are data from chamber measurement from the year before the clear cut (starting during mid-summer 2015). Without a reference-control site and only a very short time series before the clear cut (even not a full year) for CH4 and N2O, data interpretation is very difficult and weak. The weather conditions in 2015 are not discussed.

I would suggest including these data (CO2 balances before clear cut and GHG of reference/control site after clear cut) in order to get a complete dataset instead of publishing a partial data set with weakly supported main conclusion. In addition I would suggest to include basic information about soil properties and to check the paper for consistent data sets. See comments below.

I miss a discussion about the system boundaries regarding source/sink function of forests. Is the assumed sink or neutral function of the forest due to accumulation of wood? Is the source function after the clear cut due to enhanced mineralisation of logging residues (which caused the former c sink/neutral)? What's about the peat mineralisation before and after clear cutting? GHG warming potential depending on the system boundary and source/sink function may change including C export from harvest. See also (Biogeosciences, 15, 3603–3624, 2018). Hommeltenberg et al. (2014, Biogeosciences, 11, 3477–3493, 2014) stated that a forested bog was a strong carbon sink based on EC measurement – however estimation of long term carbon loss rate since drainage indicated a carbon source of the site. I would extend the discussion on clear cuts effects on GHG balances with the focus on soil type (mineral vs organic)

p 13 L21 ff. How is the carbon balance in boreal forest on mineral soils? Perhaps it is possible to get an estimation of peat carbon mineralisation after clear cutting?

Information about soil properties are missing. The only information included is the peat type (nutrient-rich peatland). Information about soil organic carbon content, storage, C/N ratio and bulk density before and after the clear cut (or of the control site) would be valuable in order to compare these results with other organic soils sites. The use of heavy machinery ( p4 L2-3) could lead to soil compaction, which could lead to a higher water table (depending on the reference point of water table measurement). Please include basic soil properties and an estimation peat thickness (reference and clear cut or before and after clearcutting).

Soil T data is not consistent in Figures. Compare Figure 2. Soil T at 5 cm depth is warmer in winter time than in summer time compared to other soil depth temperature and has the lowest annual variation (perhaps data of 30 cm depth?). In Figure S3 clear cut temperature in winter 2017 is 0°C without any variation. In Figure 2 all T soil depth are below 0°C in winter 2017. In addition there seems different sensibility of temperature sensors during the year, especially at the end of 2017 there seems temperature drops of 0.5 °C. Please clarify. Please use same axis label to be able to compare the same time periods. I would like to have the time mark of clear cut in all Figures (similar to Figure 2). Figure 5 shows all accepted night-time vs T air. Below 10°C there are very few points, below 7°C no points. However, T air seemed to be below 5°C for several months (Figure 2). The soil temperature (5cm) is below 5°C (Figure 9). How does this fit together? Does this mean that the data in Figure S2 during winter 2016/ 2017 are daytime data? In the paper is stated that data coverage is 30% after selection (p.5 l.21). Could you please add the night-time and daytime coverage (per month) of EC data?. Is the uncertainty of NEE due to uneven distribution of day-time and night-time coverage during gap-filling considered? How does Figure 9 fit to Figure 5a?

Additional comments. I wonder why CO2 fluxes from chamber measurements are

related to T soil, while EC CO2 night-time fluxes are related to T air and why T air is used in gap filling procedure. Please Comment.

The addition of annual cumulative footprint contributions in Figure 1 would help to evaluate the chosen suitable wind direction. Please add the used wind direction.

I would appreciate a table including warming potential of all GHG gasses.

Figure 3: Please include water table of control site.

―――――――――――――――――――――

---

## Referee Comment (RC2) · Anonymous Referee #2 · 4 Apr 2019

The study tested the impact of a disturbance (clearcutting) on surface greenhouse gas (CO2, CH4, and N20) fluxes in a forested boreal peatland in southern Finland. Over a 2 year period they used both eddy covariance and chamber based methods to measure GHG fluxes, as well as a number of environmental variables known to drive gas fluxes (water table, soil temperatures). The study shows a increase in CH4 and N20 emissions following disturbance, however the authors deem the CH4 to not be of importance regarding GWP. The paper is well written and contains valuable information regarding the impacts of forest management practices on the carbon balance of boreal peatlands.

My main concern is the conclusion drawn from the data presented here that clearcutting results in the forest turning in to a large CO2 source. The study includes 2 years

of CO2 flux data collected after the clearcutting and presents the trends of these 2 years well. However, there is no reference data of the CO2 balance of an undisturbed site and as such the study does not show that there has been an increase in a CO2 source effect. The study mentions unpublished data that provides this information. This data needs to be either published and referenced or including in this study for the conclusions being drawn here to be substantiated and allowed. The study site is also compared to a control, undisturbed, site where there has been water table and soil temperature collected. There is no CO2 flux data provided from this site, if data is provided from this site as reference point of the annual NEE budget of an undisturbed site in the area. A more detailed data set from this control site (including flux data as well as environmental variables, seasonal weather data) would allow for it to be used as a reference site and then an impact to the source/sink function of the site be commented upon. A map the shows the location of both sites would also be useful. I would suggest either including the unpublished data for a more complete study or a more detailed description of the control site as described above if the authors wish to maintain their conclusion that there is an increased source effect. As is presented here this conclusion can not be claimed.

My secondary concern is regards to the modelling of Reco. There is no influence of water table included in this model (Equation A3) and I would like to know what is the justification of this? The study has a good data set on water table depth throughout the years and a number of times in the paper it is mentioned that this rising water table may be influential on CO2 fluxes and suppress them. If it is being used as reasoning for low fluxes, then it should be including in the modelling attempts. Also, what was the justification to use air temperatures and not soil temperatures in the part equation A3 that is acting similar to a Q10 value?

Reco is represented well in the paper and seems to be the driving factor in annual NEE budgets that show a loss of CO2 to the atmosphere. However its is unclear what proportion of Reco is made up of Rff and CO2 lost due to enhanced mineralization of

the clearcutting leftovers/debris. Some kind of graphical representation of the contribution of these two CO2 sources to Reco would be necessary to comment on the long term impact of peat carbon following clearcutting and not just the recently felled organic matter at the surface.

Pg. 2 L. 13 nitrogen should be nitrous?

Pg. 6. L. 15 – 16 what was the justification of using a closure time of 10 – 11 mins? For N20 closure time can be very site dependent, was this tested at this site to ensure the time was enough to capture representative flux measurements?

Reporting of GPP data is not consistent in the text. Pg. 9 L. 13 – 14 GPP rates are reported using positive values. Compare this to Pg. 10 L. 10 where GPP balances are reported using negative values. I would suggest using negative values for both as this paper is addressing NEE not NEP.

---

## Author Comment (AC2) · 24 Apr 2019

The comment was uploaded in the form of a supplement:
https://www.biogeosciences-discuss.net/bg-2018-473/bg-2018-473-AC2-supplement.pdf

---

## Author Response (AR1)

**General response**

We have now studied the comments from the reviewers and revised the manuscript accordingly.

5 The most significant changes include: removing the references to the unpublished data, adding the information of soil properties, changing the statistical tests of manual chamber measurements from t-test to linear mixed effects model, and adding the chamber measurements made at the control site into the manuscript. When adding the control data into the manuscript we noticed that all the $CO_2$ chamber fluxes at the clear-cut site were calculated only with the linear regression meaning that the forest floor respiration was underestimated. This has also been corrected in the revised manuscript.

We hope the changes made into the revised version are satisfactory. See below the point-by-point response to the reviewers and the revised version of the manuscript with marked-up changes.

**Response to Reviewer #1**

**General Comments: The study investigated all three main GHG gases (CO2, CH4 and N2O) and heat fluxes after a clear cut on a moderately drained boreal organic soil. The authors used a combination of eddy-covariance and chamber method for flux determination. Studies determining the effect on GHG after clear cut exits for mineral soil, but not for organic soil. Details are nicely discussed.**

**My main concern is the experimental design of the study. The paper is based on two years of CO2 measurement after the clear cut. Reference data (fluxes before the clear cut) are missing, although it is stated in the paper, that one year measurement of CO2 fluxes before the clear cut exits. These data are not published (p.11 ll.24-25). However the main conclusion (clearcutting turned boreal forest from neutral to C source) is based on these unavailable data. In view of the generally large interannual variability of carbon balances on organic soil, the main conclusion is weakly supported. Interannual variability is not even discussed within the paper, although the first year after clear cutting is drier and warmer than the following. Without a control site including GHG measurements it is difficult to figure out the influence of the clear cut and "normal" annual variability. It exits a control plot of a reference site, but only data of water table and soil temperature is included. There is a second EC tower, measuring above the canopy (p.4 l.27). It is not clear whether GHG flux data exits and could be used as reference. There are data from chamber measurement from the year before the clear cut (starting during mid-summer 2015). Without a reference-control site and only a very short time series before the clear cut (even not a full year) for CH4 and N2O, data interpretation is very difficult and weak. The weather conditions in 2015 are not discussed. I would suggest including these data (CO2 balances before clear cut and GHG of reference/control site after clear cut) in order to get a complete dataset instead of publishing a partial data set with weakly supported main conclusion. In addition I would suggest to include basic information about soil properties and to check the paper for consistent data sets. See comments below.**

1. We understand the concern of the referee about the unpublished data. We have measured $CO_2$ exchange with the EC method at the Lettosuo site for six years before harvesting the site, and these data are fully processed and analyzed. Based on this, we can conclude with confidence that the site is on average $CO_2$ neutral. The neutrality is caused by the fact that the emission from the soil (peat decomposition) is approximately as large as the $CO_2$ uptake used for tree growth. The manuscript concerning the pre-harvest data is currently in preparation, but the expected submission/publishing date is not known yet. The original EC tower (not the one used for the submitted paper) has been measuring $CO_2$ exchange from a partially harvested area of the site since 2016, so we cannot use the 2016-2018 data as a control here. Unfortunately, we have had no EC measurements running within the actual control area either.

Because the manuscript presenting the pre-harvest data is already in preparation and including these data would make the present paper too extensive and potentially confusing,

we decided to remove the references to the unpublished data from the discussion and conclusions. We will only discuss the post-harvest balances and change the wording so that the change in $CO_2$ fluxes due to the clearcutting will not appear as a main conclusion of the paper. However, we can still draw some conclusions on the change in the $CO_2$ balance, and even in Reco and GPP, on the basis of previous studies concerning nutrient-rich peatland forests (Meyer et al., 2013; Ojanen et al., 2013; Uri et al., 2017).

2. In addition to the chamber measurements made within the clear-cut, already included in the manuscript, we have made similar measurements within the control area. We originally left the control area data out as the changes in fluxes at the clear-cut were so clear that we thought that the clear-cut data alone would have been sufficient. However, we see now why the control area measurements should be included in the paper, especially as few measurements were made before the harvest. We will include these chamber measurements from the control area into the revised manuscript.

**I miss a discussion about the system boundaries regarding source/sink function of forests. Is the assumed sink or neutral function of the forest due to accumulation of wood?**

3. We also think that the system boundaries are important when considering the overall carbon balance of the site. In this manuscript, however, we concentrated on reporting greenhouse gas fluxes between the ecosystem and the atmosphere. The measurements included in this manuscript constitute a part of a larger project, and there will also be a paper on soil hydrology and carbon export from the clear-cut area. All the results from the project will be combined into a review paper or a thesis that will provide a more complete view.

4. The pre-harvest $CO_2$ neutral state of Lettosuo is due to the significant tree growth at the site (unpublished data). Uri et al. (2017) have shown that in those well-drained peatland forests where the trees are relatively young, the trees can accumulate more carbon than is released from soil. We will add discussion on this to the revised manuscript.

**Is the source function after the clear cut due to enhanced mineralisation of logging residues (which caused the former c sink/neutral)?**

5. Even though the logging residues act as a source of $CO_2$ emissions and are a significant part of Reco (see our replies #3 and #4 to referee #2), they are not the main reason why the clear-cut acted as a source in this study. The main reason is the removal of photosynthesizing biomass (trees) and the destruction of ground vegetation. As there were no trees growing (see reply #4) and not much photosynthesizing vegetation, the GPP was low during the post-harvest summer.

**What's about the peat mineralisation before and after clear cutting? GHG warming potential depending on the system boundary and source/sink function may change including C export from harvest. See also (Biogeosciences, 15, 3603–3624, 2018). Hommeltenberg et al. (2014, Biogeosciences, 11, 3477–3493, 2014) stated that a forested bog was a strong carbon sink based on EC measurement – however estimation of long term carbon loss rate since drainage indicated a carbon source of the site. I would extend the discussion on clear cuts**

**effects on GHG balances with the focus on soil type (mineral vs organic) p 13 L21 ff. How is the carbon balance in boreal forest on mineral soils? Perhaps it is possible to get an estimation of peat carbon mineralisation after clear cutting?**

6. Based on the EC and tree growth measurements (unpublished data), we have estimated that the peat mineralisation rate at Lettosuo was about 1000 g C m-2 yr-1 before clearcutting. It is hard to estimate this rate reliably over a year after clearcutting due to mineralisation of logging residues, as they are both included in Reco. The chamber measurements, from which the peat mineralisation could be estimated, concentrated mostly around the summer, which makes the estimation of annual rates highly uncertain. As peat mineralisation should decrease with increasing water table level, we would expect decreased peat mineralisation after clearcutting. We will extend the discussion with the focus on soil type.

**Information about soil properties are missing. The only information included is the peat type (nutrient-rich peatland). Information about soil organic carbon content, storage, C/N ratio and bulk density before and after the clear cut (or of the control site) would be valuable in order to compare these results with other organic soils sites. The use of heavy machinery (p4 L2-3) could lead to soil compaction, which could lead to a higher water table (depending on the reference point of water table measurement). Please include basic soil properties and an estimation peat thickness (reference and clear cut or before and after clearcutting).**

7. Thank you for pointing this out. We will add data on soil organic carbon and nitrogen content with CN ratio and bulk density values in five different soil layers (humus, 0-10 cm, 10-20 cm, 20-30 cm, 30-40 cm). We will also add soil organic carbon and nitrogen stocks, and an estimation of peat thickness into the revised manuscript.

**Soil T data is not consistent in Figures. Compare Figure 2. Soil T at 5 cm depth is warmer in winter time than in summer time compared to other soil depth temperature and has the lowest annual variation (perhaps data of 30 cm depth?). In Figure S3 clear cut temperature in winter 2017 is 0∘C without any variation. In Figure 2 all T soil depth are below 0∘C in winter 2017. In addition there seems different sensibility of temperature sensors during the year, especially at the end of 2017 there seems temperature drops of 0.5 ∘C. Please clarify.**

8. There is a mistake in the legend of Fig. 2b, as the color scheme in the legend was shifted downwards due to a technical error. The blue color should refer to 5 cm, orange to 10 cm, green to 20 cm and red to the 30 cm temperature. We apologize for the confusion caused by this and will fix the legend.

9. The reason for the difference between Figs 2 and S3 is that they are measured with different sensors and at different places, though within 15 m from each other. The data in Fig. 2 are from the soil temperature profile measurements located at the orange dot in Fig. 1, while the temperature plotted in Fig. S3 is from the sensors located along the orange transect in Fig. 1 at 8 and 22 m from the ditch. We forgot to report the soil temperature sensors used in the transect in the methods and will add this information into the revised manuscript. Also, the resolution (0.5 °C) of the sensors located along the transect (Fig. S3) is lower than that in the profile measurement, which causes the step-like pattern in the plot.

**Please use same axis label to be able to compare the same time periods. I would like to have the time mark of clear cut in all Figures (similar to Figure 2).**

10. We will fix the axis labels and add the time mark of the clear-cut to all relevant figures (Figs. 8, S3 and S4) in the revised manuscript.

**Figure 5 shows all accepted night-time vs T air. Below 10∘C there are very few points, below 7∘C no points. However, T air seemed to be below 5∘C for several months (Figure 2). The soil temperature (5cm) is below 5∘C (Figure 9). How does this fit together? Does this mean that the data in Figure S2 during winter 2016/ 2017 are daytime data?**

11. We apologize for this, but the temperature in Fig. 5a is actually the 5 cm soil temperature (please see also reply #15) and it only includes the accepted night-time summertime data, in which the soil temperature was rarely below 10 °C. We will correct the figure label and the associated text in the revised manuscript.
Fig. 9 includes all the chamber measurement made during the respective years, so it also includes the data outside the summer period. This is why there are sub-zero temperatures in Fig. 9.
Fig. S2 includes all the accepted (both night- and daytime) half-hourly measurements made during the measurement period.

**In the paper is stated that data coverage is 30% after selection (p.5 l.21). Could you please add the night-time and daytime coverage (per month) of EC data?**

12. We will add the monthly night-time and daytime coverages to the supplement of the revised manuscript.

**Is the uncertainty of NEE due to uneven distribution of day-time and night-time coverage during gap-filling considered? How does Figure 9 fit to Figure 5a?**

13. We did not consider the uneven distribution of the data coverage in gap filling. However, this is a valid point, as the data coverage is usually much lower during the night than day and this may affect the uncertainty estimates. However, we did not consider this difference in the present study.

14. The idea of the Figs. 9 and 5a were to compare the respiration response between the years for the chamber and EC measurements, respectively. However, the time periods differ between the figures as Fig. 5a contains only the summertime EC data while Fig. 9 has all the chamber measurements made within a whole year. Therefore, one cannot directly compare the responses between these two measurement methods by using these figures.

**Additional comments. I wonder why CO2 fluxes from chamber measurements are related to T soil, while EC CO2 night-time fluxes are related to T air and why T air is used in gap filling procedure. Please Comment.**

15. We originally used air temperature for gap filling because air temperature data are better available from the surrounding weather stations. During the data analysis, however, we noticed that the time series of the soil temperature at the clear-cut was almost perfect (Fig.

2), so we decided to use that for gap filling the EC data instead. Unfortunately, we forgot to update this into the text, but this will be corrected in the revised manuscript.

**The addition of annual cumulative footprint contributions in Figure 1 would help to evaluate the chosen suitable wind direction. Please add the used wind direction.**

16. We will add into Fig. 1 windroses that show the annual footprint contributions to the accepted flux data.
17. Thank you for pointing out the missing wind directions. They will be added into the methods section in the revised manuscript.

**I would appreciate a table including warming potential of all GHG gasses.**

18. We will add the GWPs of the GHGs into the discussion text.

**Figure 3: Please include water table of control site.**

19. The time series of the water table level at the control site will be added into Fig. 3 in the revised manuscript.

**Response to Reviewer #2**

**The study tested the impact of a disturbance (clearcutting) on surface greenhouse gas (CO2, CH4, and N20) fluxes in a forested boreal peatland in southern Finland. Over a 2 year period they used both eddy covariance and chamber based methods to measure GHG fluxes, as well as a number of environmental variables known to drive gas fluxes (water table, soil temperatures). The study shows a increase in CH4 and N20 emissions following disturbance, however the authors deem the CH4 to not be of importance regarding GWP. The paper is well written and contains valuable information regarding the impacts of forest management practices on the carbon balance of boreal peatlands.**

**My main concern is the conclusion drawn from the data presented here that clearcutting results in the forest turning in to a large CO2 source. The study includes 2 years of CO2 flux data collected after the clearcutting and presents the trends of these 2 years well. However, there is no reference data of the CO2 balance of an undisturbed site and as such the study does not show that there has been an increase in a CO2 source effect. The study mentions unpublished data that provides this information. This data needs to be either published and referenced or including in this study for the conclusions being drawn here to be substantiated and allowed. The study site is also compared to a control, undisturbed, site where there has been water table and soil temperature collected. There is no CO2 flux data provided from this site, if data is provided from this site as reference point of the annual NEE budget of an undisturbed site in the area. A more detailed data set from this control site (including flux data as well as environmental variables, seasonal weather data) would allow for it to be used as a reference site and then an impact to the source/sink function of the site be commented upon. A map the shows the location of both sites would also be useful. I would suggest either including the unpublished data for a more complete study or a more detailed description of the control site as described above if the authors wish to maintain their conclusion that there is an increased source effect. As is presented here this conclusion cannot be claimed.**

1. Thank you for the comment, and we understand the problem about referencing unpublished data. Referee #1 made the same comment (please see our reply #1 to referee #1). In short, we have no reference data for $CO_2$ fluxes from the control site (which is too small for eddy covariance) and also cannot use the $CO_2$ flux data from the original EC tower as a control dataset; therefore, we have decided to only discuss the post-harvest balances and remove all the references to the unpublished data from the revised manuscript.

   We have made chamber measurements similar to the clear-cut area within the control area and will add them to the revised manuscript (please see our reply #2 to referee #1).

   The map included in the manuscript (Fig. 1) already shows the location of the control area, but we will add borders around it to make it more distinctive.

**My secondary concern is regards to the modelling of Reco. There is no influence of water table included in this model (Equation A3) and I would like to know what is the justification of this? The study has a good data set on water table depth throughout the years and a number of times in the paper it is mentioned that this rising water table may be influential on CO2 fluxes and suppress them. If it is being used as reasoning for low fluxes, then it should be including in the modelling attempts. Also, what was the justification to use air temperatures and not soil temperatures in the part equation A3 that is acting similar to a Q10 value?**

2. The referee is right that water table level affects the decomposition of organic matter and thus should affect $CO_2$ fluxes from the soil. However, it should be noted that Reco, as defined here, combines two processes: forest floor respiration (mainly soil) and the decomposition of logging residues. Water table depth affects the former but not the latter, which is mostly controlled by temperature. According to our chamber measurements (with no above-ground residues), the $CO_2$ emissions from the forest floor were 32% (2016) and 57% (2017) of Reco (P. 12, L. 11-17), meaning that the logging residues contribute markedly to Reco. One should also take into account that the chamber measurements also include below-ground residues (e.g. roots), so in reality the contribution of the residues to Reco is even larger.

We plotted residuals of the respiration fit (Eq. A3) against WTL for both summers (Fig. 1) and found no significant relationship between respiration and WTL. This is discussed in the original manuscript (P. 13, L. 9-14). Thus we did not include WTL as an explanatory variable in the Reco model. We will add a comment on this and the residual-vs-WTL plot to the revised version.

3. Regarding the temperature issue, please see our response to Referee #1 (reply #15). In short, we actually did use soil temperature in Eq. A3, but forgot to update this into the text and figure captions. We apologize for confusion and will fix this in the revised manuscript.

[Figure]

Figure 1. Hourly mean residual of the respiration fit vs. hourly mean water table level in Jun-Aug 2016 (blue) and 2017 (orange).

**Reco is represented well in the paper and seems to be the driving factor in annual NEE budgets that show a loss of CO2 to the atmosphere. However it's is unclear what proportion of Reco is made up of Rff and CO2 lost due to enhanced mineralization of the clearcutting leftovers/debris. Some kind of graphical representation of the contribution of these two CO2 sources to Reco would be necessary to comment on the long-term impact of peat carbon following clearcutting and not just the recently felled organic matter at the surface.**

4. As indicated above (reply #2), the $CO_2$ emissions from the forest floor were 32-57% of Reco, while the logging residues probably account for the rest. The aim of our study was to investigate the short-term impacts of clearcutting, and discussion of any long-term impacts would be mostly speculative. Likely, the $CO_2$ emissions from the logging residues will decrease in the next few years as they decompose. Also, the planted trees and the ground vegetation will grow; therefore, the transpiration will increase causing the WTL to decrease. This will again likely increase $CO_2$ emission from the soil to the same level as before the clearcutting at some point. Anyhow, our focus is on the $CO_2$ emissions right after the clear-cut, so we would prefer not to add any further discussion on longer-term effects.

The idea of graphical presentation is great, and we will add a bar plot to the revised manuscript, which shows the distribution of estimated Reco to $R_{residues}$ and Rff for both summers.

**Pg. 2 L. 13 nitrogen should be nitrous?**

    5. Yes you're right. This will be corrected in the revised manuscript.

**Pg. 6. L. 15 – 16 what was the justification of using a closure time of 10 – 11 mins? For N20 closure time can be very site dependent, was this tested at this site to ensure the time was enough to capture representative flux measurements?**

    6. Yes, we tested the chamber system at our site before the clearcutting in 2015 with a longer closure time (20 min) and noticed that by selecting only the first 10 min of the data gave very similar results to using the whole 20 min dataset. Because of this, we decided to limit the closure time to 10-11 min to allow more spatial replicates to be measured during the same day. After clearcutting, the $N_2O$ fluxes increased so much that we could have actually reduced the closure time down to 5 min. However, due to the fact that the absolute $CH_4$ flux decreased (from sink to small source) after clearcutting, we still had to use a closure time of 10-11 min to measure the $CH_4$ fluxes with sufficient accuracy. Also, keeping the closure time the same allowed a more consistent comparison between the years.

**Reporting of GPP data is not consistent in the text. Pg. 9 L. 13 – 14 GPP rates are reported using positive values. Compare this to Pg. 10 L. 10 where GPP balances are reported using negative values. I would suggest using negative values for both as this paper is addressing NEE not NEP.**

    7. Thank you for pointing this out. We decided to use the absolute value of GPP (|GPP|) in the text to avoid the confusion that occurs commonly when talking about increasing/decreasing GPP. However, we accidentally left some negative values in one section, as pointed out by the referee. The values in that section will be changed from negative to positive in the revised manuscript.

[revised manuscript text omitted]

---

## Referee Report (RR1)

Review of Korkiakoski et al

**Greenhouse gas and energy fluxes in a boreal peatland forest after clearcutting**

General comment: This is a great data-rich paper on a highly relevant timely topic on GHG fluxes from peatlands following clear cutting. The authors made a great effort in writing and revising their manuscript. They have clearly addressed the reviewer comment which has greatly improved the manuscript.

I only have minor technical issues that are easily to address.

1. I find 'forest floor respiration' somewhat misleading as respired CO2 originates also from deeper soil layer - consider to rename it to soil respiration or soil CO2 efflux.

2. Statistical analyses: please define the 'fixed' effects.

3. at p. 19 L. 3-10, the reporting of long-term temperatures is somewhat confusing as you first report temperatures measured at the climate station, then yours and then the ones of the climate station again. Please try to present this more coherent.

4. at p. 25. L.19. 'This also means…' is an awkward sentence. Please improve.

5. Table 1: Soil properties. Is the classification as ‚humus layer' correct ? The extremely low density would rather suggest that it is a 'litter layer' that is also more clearly distinguishable from peat. What is the thickness?

---

## Author Response (AR2)

Dear editor,

We have now made all the technical corrections suggested by the referees except the one (#1) below which we disagree with. On behalf of the authors, I would like to thank you for taking the time to review this manuscript.

5    Referee #2:

**1. I find 'forest floor respiration' somewhat misleading as respired CO2 originates also from deeper soil layer - consider to rename it to soil respiration or soil CO2 efflux.**

We disagree because the respiration we measured did not only include soil respiration but also plant respiration. We can't use soil CO2 efflux either as it by definition does not include the above-ground parts of the plants, only the roots. We think that
10   'forest floor respiration', which we defined as the sum of autotrophic and heterotrophic respiration (Pg. 7 L. 21-22), is the best definition to use in our case.

**5. Table 1: Soil properties. Is the classification as 'humus layer' correct ? The extremely low density would rather suggest that it is a 'litter layer' that is also more clearly distinguishable from peat. What is the thickness?**
15   Yes, it should be litter layer and it's now corrected. Unfortunately, the litter layer thickness was not recorded.

Best regards,
Mika Korkiakoski

[revised manuscript text omitted]
 fixed effects of the model were the site type (clear-cut or control) and the measurement year when comparing the measurements made at different sites and in different years, respectively. The linear mixed effect model was carried out with the R programming language (R Core Team, 2018, version 3.5.0) using the 'lme4' package. The normality of model residuals was visually checked using quantile-quantile plots. The differences were tested with Tukey's HSD post hoc test.

The analysis of the clearcutting effects on WTL was based on the paired treatment approach (also called calibration period – control area method) (e.g., Kaila et al., 2014; Laurén et al., 2009). We first calculated linear regressions between the WTL within the control and clear-cut sites for the pre-treatment period using the WTL logger data from 2015. Then we used this regression model and the post-treatment WTL data from the control site to predict WTL for the clear-cut site as if it had not been harvested. The clear-cut effect was calculated as the difference between the calibrated post-clear-cut WTL measurements and the predicted background WTL values in the clear-cut site after the harvest.

**2.7 Water table level measurements and analysis**

The water table levels within both the clear-cut and non-managed control site were measured for one year before (2015) and two years after the clearcutting (2016-2017). Four automatic monitoring plots consisting of dipwells (perforated plastic tubes 120 cm long, 3.5 cm diameter) were set up at the centre of each site (Fig. 1), located at a distance of 4, 8, 12 and 22.5 m in a transect perpendicular to the ditch (ditch spacing was 45 m). In addition, in order to calibrate the automatic water table measurement data, manually monitored dipwells were installed close to the automatically monitored dipwells within the clear-cut and control sites. WTL was measured manually at weekly or fortnightly intervals during March–November. From the automated dipwells, WTL was recorded with automatic probes (TruTrack WT-HR -logger, Intech Instruments Ltd, Auckland, New Zealand; Odyssey Capacitance Water Level Logger, Dataflow Systems Limited, Christchurch, New Zealand) at hourly

intervals. The recorded values were then calibrated with linear regression using the manually measured WTL data from both the control and clear-cut site.

The analysis of the clearcutting effects on WTL was based on the paired treatment approach (also called calibration period – control area method) (e.g., Kaila et al., 2014; Laurén et al., 2009). We first calculated linear regressions between the WTL within the control and clear-cut sites for the pre-treatment period using the WTL logger data from 2015. Then we used this regression model and the post-treatment WTL data from the control site to predict WTL for the clear-cut site as if it had not been harvested. The clear-cut effect was calculated as the difference between the calibrated post-clear-cut WTL measurements and the predicted background WTL values in the clear-cut site after the harvest.

**3 Results**

**3.1 Meteorological and hydrological conditions**

The long-term (1981–2010) mean annual, winter (DJF), and summer (JJA) air temperatures at the nearby (Jokioinen, 35 km northwest of Lettosuo) weather station were 4.6, –5.3, and 15.2 °C, respectively (Pirinen et al., 2012). The mean annual temperature before the clearcutting in 2015 at Jokioinen was 6.2 °C. Also, the winter (2015–2016) before the clearcutting was warmer (–3.4 °C) while the summer 2015 was colder (14.4 °C) than the long-term mean. The mean post-clearcut annual air temperatures at the EC 
[revised manuscript text omitted]

*Author contributions*. TP, PO, KM, TL and AL designed the study. Field measurements and maintenance were carried out by MK, JR, TL, and AL. J-PT made spectral corrections and footprint analysis for the EC data. SS and PO corrected the WTL data and did the WTL background modelling. Rest of the data analysis was carried out by MK. MK wrote the paper with contributions from all coauthors.

*Competing interests*. The authors declare that they have no conflict of interest.

*Acknowledgements*. We are grateful for the financial support from the Maj and Tor Nessling foundation and from the Ministry of Transport and Communications through the Integrated Carbon Observing System (ICOS) research.

[revised manuscript text omitted]